# Unaltered hepatic wound healing response in male rats with ancestral liver injury

Johanna Beil[1,3], Juliane Perner[1,3], Lena Pfaller[1,3], Marie-Apolline Gérard[1], Alessandro Piaia[1], Arno Doelemeyer [1], Adi Wasserkrug Naor[2], Lori Martin[2], Aline Piequet[1], Valérie Dubost[1], Salah-Dine Chibout[1], Jonathan Moggs[1] & Rémi Terranova [1] ✉

The possibility that ancestral environmental exposure could result in adaptive inherited effects in mammals has been long debated. Numerous rodent models of transgenerational responses to various environmental factors have been published but due to technical, operational and resource burden, most still await independent confirmation. A previous study reported multi-generational epigenetic adaptation of the hepatic wound healing response upon exposure to the hepatotoxicant carbon tetrachloride ($CCl_4$) in male rats. Here, we comprehensively investigate the transgenerational effects by repeating the original $CCl_4$ multigenerational study with increased power, pedigree tracing, F2 dose-response and suitable randomization schemes. Detailed pathology evaluations do not support adaptive phenotypic suppression of the hepatic wound healing response or a greater fitness of F2 animals with ancestral liver injury exposure. However, transcriptomic analyses identified genes whose expression correlates with ancestral liver injury, although the biological relevance of this apparent transgenerational transmission at the molecular level remains to be determined. This work overall highlights the need for independent evaluation of transgenerational epigenetic inheritance paradigms in mammals.

In recent years, multiple cross-generational exposure paradigms, including epidemiological studies and animal models, have explored the idea that ancestral environmental experiences (e.g., related to parental diet, traumatic experiences, toxin exposure) may be written into our epigenomes and transmitted through the germline to influence development and health of progeny[1–5]. This would have profound implications for the understanding of human health and disease[6,7]. However, the role and nature of germline epigenetic effects in phenotypic transmission is disputed[8–10] and the overall relevance and penetrance of the transgenerational epigenetic inheritance (TEI) phenomenon in mammals remains uncertain[11–13]. Specifically, many of the studies do not represent bona fide cases of TEI, but rather inter-generational effects where germline exposure or other confounding

effects (e.g., postnatal nutritional environment, behavioral effects, microbiotic effects, metabolites, or cryptic genetic variations) might contribute to the transmitted phenotypes. Furthermore, the repro-gramming of the epigenome in primordial germ cells during gameto-genesis and following fertilization represent major barriers to TEI in mammals[14]. Yet the transmission of epigenetic changes cannot be excluded, as illustrated by imprinted genome loci that resist waves of epigenome reprogramming post-fertilization[15]. Thus, at present TEI is plausible, but much work remains to be done to establish strong evidence of this phenomenon in mammals[13,16,17].

To evaluate the existence of TEI, it is important to assess non-mendelian transmission of complex pathophysiological traits using in vivo models, in which the study design accounts for and mitigates

[1]Novartis, Biomedical Research, Basel, Switzerland. [2]Novartis, Biomedical Research, East-Hanover, NJ, USA. [3]These authors contributed equally: Johanna Beil, Juliane Perner, Lena Pfaller. ✉e-mail: remi.terranova@novartis.com

important biasing factors. In 2012, a study reported multigenerational epigenetic adaptation of the hepatic wound healing response[18]. Using outbred adult male Sprague Dawley (SD) rats and repeated exposure to the hepatotoxin carbon tetrachloride (CCl₄), in a multigeneration male-transmission setting, the study highlighted that ancestral liver damage could lead to heritable reprogramming of hepatic wound healing and greater fitness of subsequent generation's exposure to the liver fibrosis inducing agent. The observed phenotypic adaptation was characterized by reduced liver myofibroblasts, increased expression of anti-fibrogenic and reduced expression of pro-fibrogenic factors.

Interestingly, CCl₄ injury recapitulates the progressive stages of human fatty liver disease, from simple steatosis, to inflammation, fibrosis, and cancer[19] and triggers similar cellular and molecular events in both human and model organisms[20]. CCl₄-induced acute liver injury thus represents a translationally relevant model of liver fibrosis, and the original study by Zeybel et al. represented an important advance in studying multigenerational epigenetic adaptation of the hepatic wound healing response.

Given the elusive evidence for TEI in mammals and the multiple effects that may confound the experimental evaluation and interpretation of this phenomenon[17,21], we aimed to comprehensively re-evaluate the potential for transgenerational adaptation of the hepatic wound healing response in a de novo multigeneration CCl₄ study in outbred SD rats. We expanded on the original work[18] by considering additional study parameters that mitigate potentially confounding effects, including standardized housing, care, pedigree tracing, study power, F2 dose–response evaluation, careful randomization schemes, and orthogonal validation of key hepatic transcriptomic findings. Based on detailed microscopic and clinical pathology evidence, our data do not support an adaptive phenotypic suppression of the hepatic wound healing response to CCl₄ as there was no evidence of reduced liver fibrogenesis or greater fitness of animals with ancestral liver injury. However, using a comprehensive RNA sequencing-based evaluation of the hepatic transcriptome, and orthogonal validation of key gene expression changes, we identify a biologically relevant set of genes whose expression pattern correlates with ancestral CCl₄-induced liver injury, suggesting transgenerational transmission at the molecular level. In the absence of detectable phenotypic effects, the functional relevance of this hepatic transcriptional programming or associated transmission mechanisms is unclear. Overall, our work emphasizes the need for independent and extended evaluation of transgenerational epigenetic inheritance paradigms in mammals.

## Results

### A multigenerational male Sprague Dawley rat study to evaluate liver adaptation phenomenology

In this study, we aimed at independently evaluating the multigenerational epigenetic adaptation phenomenon reported in 2012[18]. Our study was designed to both reproduce and expand on the original publication. We performed a multigenerational study based on a block-design including CCl₄ or vehicle control (olive oil) treatment groups in outbred adult male SD rats at generations F0, F1 and F2 (Fig. 1a). To induce a state of chronic wound healing leading to fibrosis, F0 and F1 male groups received 50% CCl₄ doses three times weekly (t.i.w.) via oral gavage for 6 continuous weeks. Animals were allowed to recover for 2 weeks prior to mating with uninjured females. For F2 phenotypic characterization, dose–response treatment (vehicle, 8% CCl₄ and 50% CCl₄) was performed. F2 animals were sacrificed, and samples were collected 24 h after the final treatment (vehicle or CCl₄). In this design, and as in the Zeybel et al. study, four distinct cohorts of ancestral liver injury are available: cohort A with no ancestral exposure to CCl₄-induced liver injury, cohort B with parental F1 injury, cohort C with grand-parental F0 injury and cohort D with both parental F1 and grand-parental F0 injury (Fig. 1a).

Multiple potential confounding effects (experimental and/or biological) may contribute to the phenotypic variations and influence the outcome and interpretation of such complex and long-lasting multigeneration studies[17,21]. We thus designed and ran the study considering the following specific study features: 1. We ensured standardized housing and care conditions for all groups through the study. 2. The study was well powered with $n = 12$ animals in each F0 and F1 group (transmission) and $n = 10$ animals in F2 groups (phenotypic evaluation). 3. To manage possible inter-individual genetic variations effects, we ensured F0 and F1 pedigree representation for all F2 animals and provide full individual animal assignment data (schematized Fig. 1b, detailed in Supplementary Methods). 4. Three dose groups were also evaluated in each F2 cohort to enable phenotypic evaluation of baseline (vehicle) and dose–responsive effects (low-dose 8% CCl₄ and high-dose 50% CCl₄) 24 h after last dose. 5. Given the large number of F2 animals (4 ancestral cohorts and 3 treatment groups, $n = 120$ in total) and to avoid treatment or collection biases, we used a carefully designed staggering and randomization scheme for F2 dose–response treatment, collection, and evaluation. 6. We carefully monitored and sampled animals throughout the study to enable comprehensive phenotypic evaluation in all three generations (illustrated in Fig. 1c and detailed in "Methods" and Supplementary Methods).

### CCl₄ treatment leads to expected and consistent liver injury across generations

To evaluate whether 6 weeks t.i.w. CCl₄ oral gavage led to the expected and consistent liver fibrosis, we evaluated clinical signs, serum biochemistry and liver samples for histopathology across generational treatments. Clinical observations, body weight and food consumption were collected throughout the study indicating treatment-related signs of toxicity associated with administration of CCl₄ as detailed in Supplementary Data 1. In F0 and F1, at high-dose (50%) CCl₄ treatment, biochemistry analysis of serum sampled 24 h after last dose showed up to marked elevation of the serum liver enzymes aspartate aminotransferase (AST, Fig. 1d, f), alanine aminotransferase (ALT, Fig. 1e, g), alkaline phosphatase (ALP) activities, as well as an increase in total bilirubin (BILI, both direct and indirect) and a minimal increase in urea (UREA). Upon recovery in pre- and post-mating samples, serum liver enzyme activities returned towards control values (Supplementary Fig. 1).

F2 animals were evaluated 24 h after final treatment of vehicle, low-dose (8%) and high-dose (50%) CCl₄ treatment. Focusing on ancestrally naive cohort A animals (groups 11, 12, 13), we found dose-dependent elevation in liver enzymes (AST, ALT, ALP) activities and increase in total bilirubin. Additionally, Sirius Red staining of FFPE liver sections revealed a dose-dependent increase in collagen deposition (Fig. 1h–k, Supplementary Fig. 3). Testicular toxicity had been reported as a consequence of oxidative stress (OS) state, following CCl₄ exposure[22–24] and could influence interpretation of our results with respect to existence of TEI. No gross testis abnormalities were detected upon necropsy of CCl₄-treated animals 24 h after last dose. In addition, no significant changes in sperm count or testis weight were observed from the testes of F2 (Supplementary Fig. 2). In summary, CCl₄ treatment leads to expected and consistent liver injury across generations and in the absence of detectable testis effects. This new study enables robust de novo evaluation of ancestral liver fibrosis influence of the hepatic wound healing response.

### Absence of microscopic or clinical pathology-based evidence for multigenerational adaptation of the hepatic wound healing response

Sirius Red staining of FFPE liver sections was used to evaluate deposition of fibrotic collagens (type I and III) in F2 animals. The primary aim of this evaluation was to investigate whether high-dose group animals with ancestral exposure (groups 16, 19 and 22) may

show lower, ancestral exposure-dependent, amounts of fibrotic matrix as compared to ancestral naive animals (group 13) as reported[18]. With some inter-individual variability, we observed a comparable percentage of the total area stained with Sirius Red across groups 13, 16, 19, and 22, regardless of ancestral liver injury exposure (Fig. 2a, right panel, Fig. 2b). Likewise, we could not detect lower gene expression of hepatic collagen I (*Col1a1* and *Col1a2*) or other liver fibrosis-promoting genes such as *Lox*[25] or *Timp1*[26] in any of the dose groups evaluated (Fig. 2c).

We hypothesized that discrete multigenerational effects might be overshadowed by the strong fibrotic response in the high-dose

treatment groups. As part of our study design, we included vehicle and low-dose treatment in F2 to evaluate potential effects in conditions of low to no fibrogenic response. Looking at animals within each cohort A–D, Sirius Red evaluation of vehicle and low-dose treatment groups showed the expected dose-responsive increase in Sirius Red positive area (Supplementary Fig. 3a). However, across cohorts there was no evidence for ancestral effects (Fig. 2a).

To ensure thorough evaluation of potential liver adaptation effects, we additionally evaluated H&E-stained liver slices for standard histopathology evaluation. Histopathology findings included centrilobular fibrosis, hepatocyte vacuolation, hepatocellular

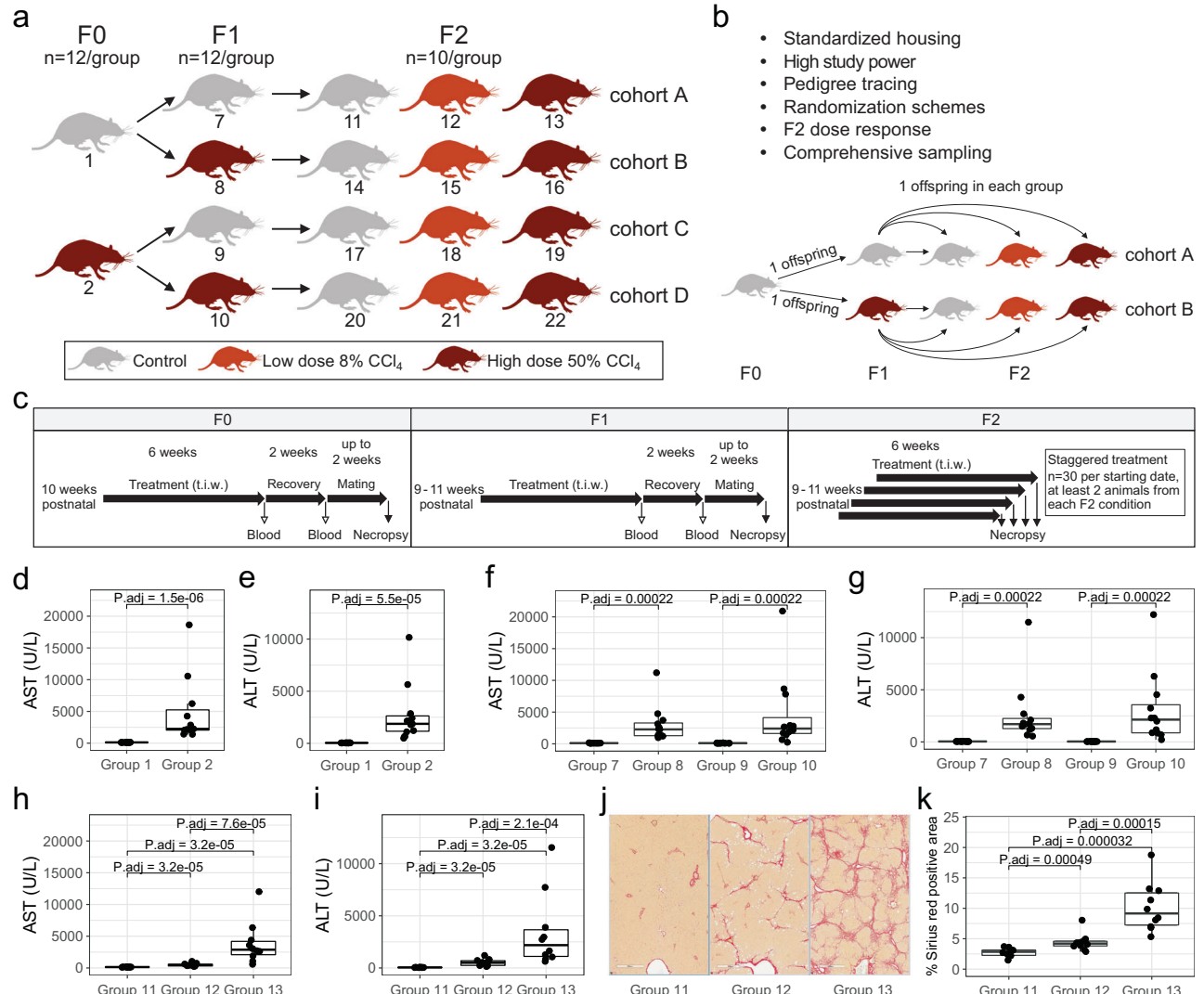

**Fig. 1 | Enhanced study for evaluating ancestral influence of CCl₄-induced liver injury. a** Multigenerational Sprague Dawley male rats CCl₄ study scheme. The group numbers (1–22 from F0 to F2) and number (*n*) of animals per generation/dose group across cohorts A, B, C and D are indicated. **b** List of study design elements addressing potential confounding effects accounted for in this study and schematic overview of the F0–F2 pedigree tracing (detailed in Supplementary Methods). **c** Illustrative F0, F1 and F2 in life study scheme. The animal age at treatment start, the length of treatment, of recovery, and of mating periods, as well as blood or necropsy timepoints are indicated. t.i.w: three times weekly treatment. **d, e** Box plots showing serum biochemistry-based AST and ALT levels (U/L, units per liter) 24 h after last dose for each F0 animal groups 1 (*n* = 12) and 2 (*n* = 11). Additional timepoints and biochemistry markers data in Supplementary Fig. 1a. **f, g** Box plots showing AST and ALT levels for F1 groups 7–10 (*n* = 12) 24 h after last dose. Additional timepoints and biochemistry markers data in Supplementary

Fig. 1b. **h, i** AST and ALT levels in ancestrally naive cohort A F2 animals (groups 11–13, *n* = 10). The values for all measured biochemistry markers are available in Supplementary Data 3, extended clinical observations and (clinical) pathology information is provided in Supplementary Data 1. **j** Representative images of Sirius Red (collagen deposition) stained livers 24 h after last dose from one animal of each ancestral naive cohort A groups 11–13 (scale bar 500 µm). **k** Quantification of Sirius Red positive area in percent of total area (*y*-axis) for each cohort A F2 animal group (*n* = 10). Two-sided Wilcoxon rank sum test was used for indicated pairwise comparisons, the individual *p* values are shown. For all box plots, the median (central line) and the lower and upper quartiles (box limits) are displayed. Whiskers extend to the maximal and minimal value or, if exceeded, to max. the 1.5 × inter-quartile range. Black points represent individual animal values of one sample per animal. Source data are provided as a Source Data file.

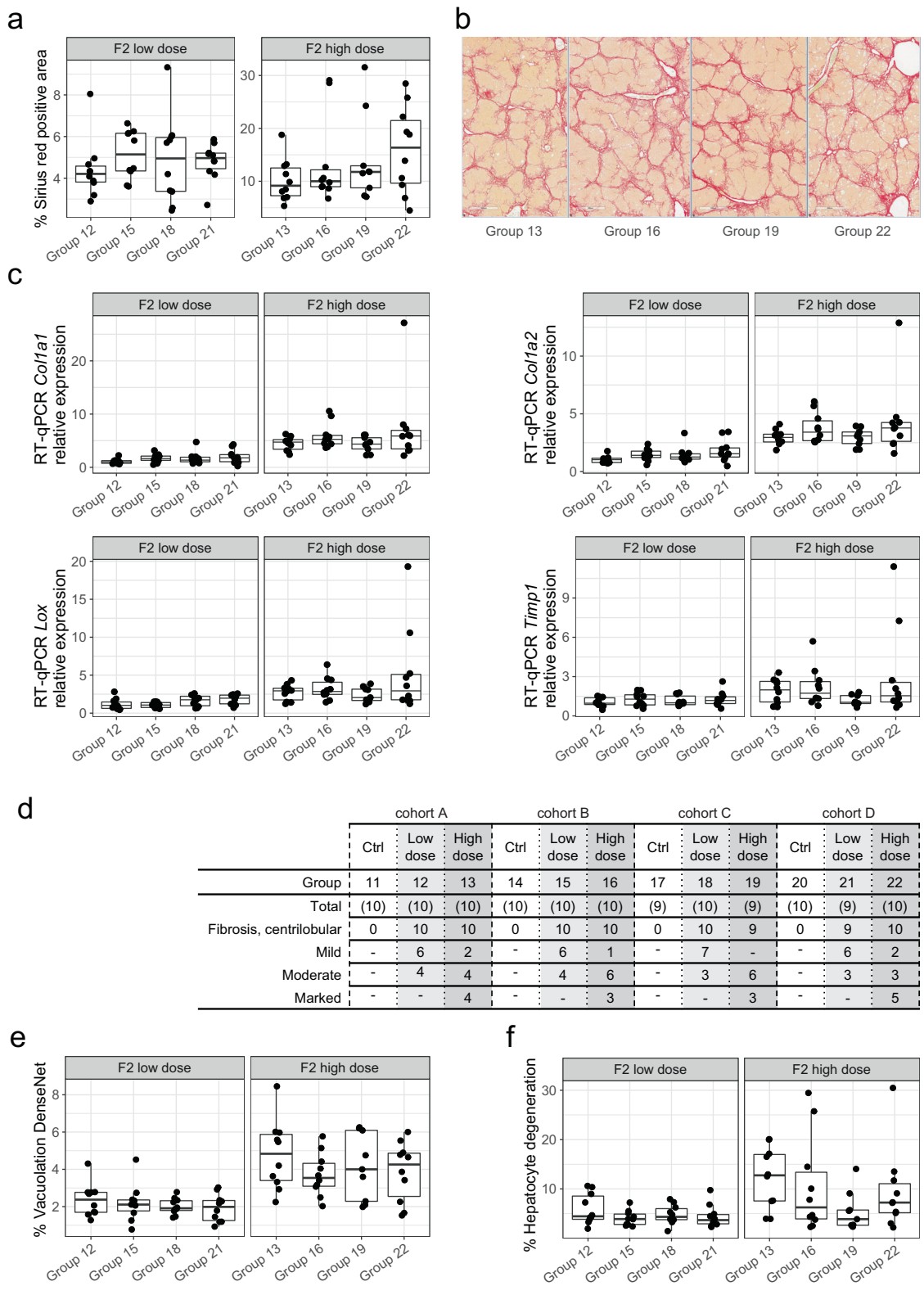

degeneration/necrosis, hepatocellular regenerative hyperplasia, hepatocyte karyomegaly, hepatocyte altered cellular foci, biliary hyperplasia, and pigment deposits. High-dose (50%) $CCl_4$ treated groups 13, 16, 19, and 22 showed a combination of severe liver changes (degenerative, fibrotic and hyperplastic), affecting all animals in a relatively similar manner regardless of ancestral F0 or F1 liver injury (Fig. 2d). Only for hepatocyte degeneration/necrosis, rats from group 13 appeared to be slightly more affected than the remaining $CCl_4$-

treated groups, in terms of severity (Supplementary Data 1). Low-dose $CCl_4$ treated groups 12, 15, 18 and 21 on the other hand showed mild liver changes (degenerative and fibrotic), affecting in similar manner all animals, regardless of the ancestral $CCl_4$ exposure. Quantitative computational image analysis (see "Methods") was further applied to quantify areas of liver vacuolation and hepatocyte degeneration. No ancestral exposure effects were observed for liver vacuolation at either low or high-dose $CCl_4$ treatment (Fig. 2e). Coherent with

**Fig. 2 | The presence of liver injury in male ancestors does not reduce liver fibrogenesis in F2 generation male offspring. a** Quantification by HALO analysis of Sirius Red positive area in percent of total area (*y*-axis) for low-dose (left) and high-dose (right) treated F2 animals by animal groups from all ancestral liver injury cohorts (*x*-axis). Pairwise, two-sided Wilcoxon rank sum tests showed no significant differences (Holm's method adjusted *p* value > 0.05) between animal groups of the same F2 dose group. **b** Representative images of Sirius Red stained livers 24 h after last dose from one animal of each high-dose treated group 13, 16, 19, 22 (scale bar 500 μm). **c** RT-qPCR-based relative mRNA expression levels (*y*-axis) of selected fibrogenic genes: Collagen Type I (*Col1a1* and *Col1a2*), Lysyl Oxidase (*Lox*) and TIMP Metallopeptidase Inhibitor 1 (*Timp1*). Relative expression was calculated by ΔΔCT with *Hprt1* as control gene and normalized to mean of cohort A low-dose group 12. Data for each gene is shown by animal group (*x*-axis) and split into F2 low-dose and high-dose groups. A Kruskal–Wallis rank sum test performed per dose group indicated no significant difference among the animal groups within each dose

group (*p* value > 0.05). **d** H&E based histopathological grading of periportal fibrosis in control, low- and high-dose animals from all animal groups (row "Group") across ancestral cohorts. The number of animals evaluated in each group (row "Total"), of animals showing centrilobular fibrosis, and the histopathology grade subset are indicated ("-": no finding). Quantitative computational image evaluation (DenseNet, see "Methods") of percentage of vacuolation (**e**) and of hepatocyte degeneration (**f**) in total area (*y*-axis). Data is presented by animal groups for low- and high-dose treated F2 animals. In all panels containing box plots, statistics are based on *n* = 10 animals for each group, except for group 19 where *n* = 9. Kruskal–Wallis rank sum test performed on each dose group indicated no significant difference among the animal groups within each dose group (*p* value > 0.1). For all box plots, the median (central line) and the lower and upper quartiles (box limits) are displayed. Whiskers extend to the maximal and minimal value or, if exceeded, to max. the 1.5 × inter-quartile range. Black points represent individual animal values of one sample per animal. Source data are provided as a Source Data file.

histopathological microscopic evaluation, rats from high-dose group 13 displayed slightly increased levels of hepatocyte degeneration (Fig. 2f). In this evaluation, no significant (p.adj > 0.05) ancestral effect was observed in low-dose animals.

To further confirm the histological fibrosis and orthogonally evaluate potential cross-generational effects, we next assessed alpha-smooth muscle actin (αSMA), a marker of myofibroblasts, which are important drivers of liver fibrosis. Anti-αSMA IHC staining of FFPE liver sections revealed a dose-dependent increase in myofibroblasts in all generational cohorts (Fig. 3a, c). No apparent ancestral exposure effects were observed in either dose group as illustrated in Fig. 3b and quantified in Fig. 3c. Likewise, the expression of the αSMA encoding gene *Acta2* while influenced by treatment did not further detect ancestral influence for this fibrotic marker (Fig. 3d). Finally, in all groups, $CCl_4$ treatment-related changes were dose-related and included marked increase in AST, ALT and ALP activities. No clear cross-generation related changes were seen in any evaluated clinical pathology parameter (Fig. 3e, f, Supplementary Fig. 3b). In summary, based on quantitative pathology and multi-parametric evaluation of key hepatic wound healing response features, no clear influence on ancestral liver fibrosis injury was detected in this independent multi-generational study evaluation.

## $CCl_4$ treatment leads to extensive and dose-responsive changes in gene expression in F2 rat livers

Genome-wide transcriptomic and proteomic analysis of $CCl_4$ treated rat liver samples have highlighted vast changes in gene and protein expression underlying liver lesions and fibrosis, pointing to multiple biological processes and molecular pathways responding to chronic $CCl_4$ treatment[20]. Thus, genome-wide, quantitative, and unbiased approaches, such as RNA sequencing (RNAseq), may provide additional molecular pathology information to explore potential multi-generational effects in more depth. We thus investigated the F2 liver samples for gene expression changes using RNAseq, randomly selecting 7 out of 10 animals from each cohort (A to D) and treatment group (11–22) for profiling.

First, we evaluated dose–response related transcriptional effects to ensure we recapitulated reported $CCl_4$ treatment associated gene expression changes. The analysis is exemplified in the following paragraph based on ancestrally naive cohort A. In the low-dose group 12 vs control group 11 comparison, 708 genes were differentially expressed (FDR < 0.1 and absolute log2 fold-change > 1) with predominately increased expression (93% of the differentially expressed genes, Fig. 4a left panel). In the high-dose vs control comparison, 3965 genes were found differentially expressed (FDR < 0.1 and absolute log2 fold-change > 1, Fig. 4a right panel) again leaning towards upregulated gene expression (71% of the differentially expressed genes). Gene ontology (GO) over-representation analysis on the differentially expressed genes for each comparison separately, showed enrichment

for genes related to extra-cellular matrix organization, chemotaxis, and wound healing for both, low-dose and high-dose contrasts (Fig. 4b). Differentially expressed genes in the low-dose group showed enrichment for genes related to cell division and tissue regeneration, while differentially expressed genes in the high-dose group showed enrichment for metabolic processes, related to fatty acids, steroids and chemicals. Focusing on known fibrosis markers and on dose–response effects, we found the expected increase in *Col1a1*, *Acta2*, and *Tgfb1* gene expression, and the expected decrease in *Pparg* gene expression (Fig. 4c).

To evaluate similarity in gene expression profiles globally, we explored all F2 liver samples in a Principal Component Analysis (PCA) based on the top 3000 most variable genes across all samples. The main principal component PC1 represents 56% of the variation among the gene expression profiles, dominating over the remaining PCs (each explaining ≤ 5% of variation). PC1 captures the dose-dependent treatment effect, separating the samples independent of ancestral background (Fig. 4d). Additionally, we repeated the differential gene expression analysis described above also for cohorts B, C and D, applying the same FDR and absolute log2 fold-change for each comparison (Supplementary Fig. 4a–h). Comparison of the selected differentially expressed genes showed general overlap in molecular changes (Fig. 4e) and of enriched GO-terms across cohort A–D treatment groups (Supplementary Fig. 4i), including biological functions related to extra-cellular matrix, cellular proliferation (e.g., cell division, nuclear division), and inflammation (e.g. leukocyte chemotaxis, cytokine production and T-cell activation). Thus, no gross differences in liver injury and associated proliferative or inflammatory response were detected. Taken together, these results show that $CCl_4$ treatment leads to expected, consistent and dose-responsive liver transcriptional perturbations in F2 animals and across ancestral cohorts A–D.

## Transcriptional analyses identify genes whose expression correlates with ancestral liver damage

We next evaluated whether ancestral liver injury may influence gene expression patterns within each treatment group. We first explored each treatment group across cohorts A–D in a separate PCA (Fig. 5a–c). As expected, we found that variation in gene expression profiles captured by each PC is relatively low (e.g., 12%, 10%, and 35% for PC1 on control, low-dose, and high-dose samples, respectively). However, for F2 control groups, PC2 showed a separation of group 11 from all samples with ancestral injury (14, 17, and 20) (Fig. 5a). We observed a similar effect in the PCA with the low-dose groups, where PC1 separates group 12 from the remaining groups (Fig. 5b). No obvious separation on the first two PCs was observed in the high-dose groups (Fig. 5c), possibly due to the dominance of the fibrosis phenotype (Fig. 4).

To identify specific genes whose expression pattern correlates with ancestral liver injury, we next modeled the transcriptional gene expression response based on F0, F1 and F2 treatment status of each

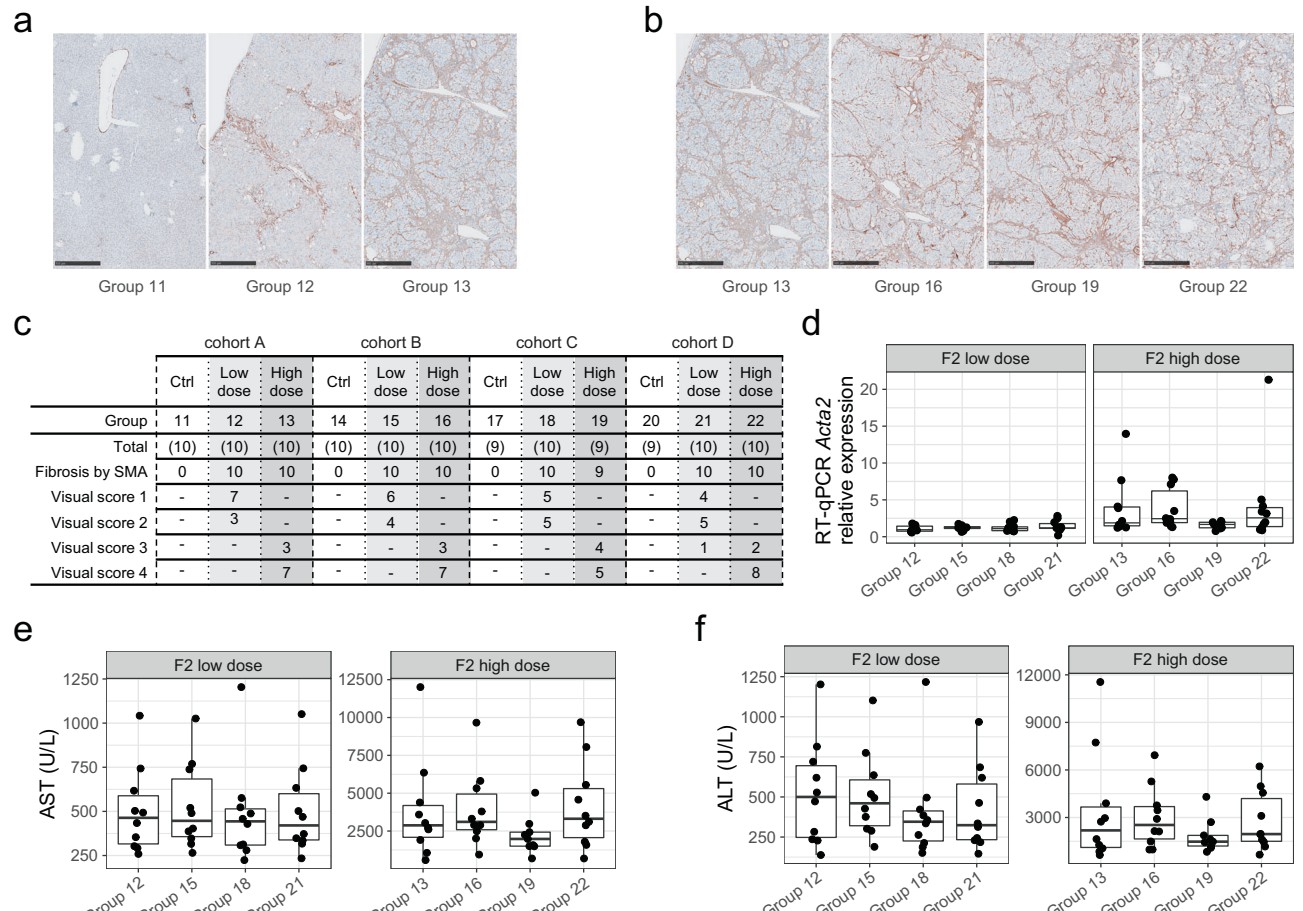

**Fig. 3 | Unaltered hepatic injury and myofibroblast activation in F2 animals with ancestral liver injury. a–c** anti-αSMA based IHC evaluation of hepatic myofibroblasts based on representative images of anti-αSMA-stained livers 24 h after last dose. **a** One representative image for each of the cohort A groups 11–13 (scale bar 500 μm). **b** One representative image for each high-dose group (13, 16, 19, 22) from all four ancestral cohorts (scale bar 500 μm). **c** Summary of the αSMA enrichment grade after visual quantification for all animal groups (row "Group") across ancestral cohorts. Row 'Total': number of animals evaluated in each group; Row 'Fibrosis by SMA': number of animals showing fibrosis based on anti-αSMA staining. Rows 'visual score': Subset according to visual grade 1 to 4 ("-": no findings). **d** RT-qPCR based relative mRNA expression evaluation of αSMA encoding gene *Acta2* from all F2 low-dose and high-dose treated samples by indicated animal group. Relative expression (*y*-axis) was calculated by ΔΔCT with *Hprt1* as control

gene and normalized to mean of cohort A low-dose group 12. **e, f** Serum biochemistry-based evaluation of AST and ALT levels (*y*-axis, U/L, units per liter) are shown for F2 animals by animal group (*x*-axis). For better and direct cross-treatment and pathology readout comparability, the data from individual dose groups (left: low-dose 8% and right: high-dose 50% CCl₄) are shown with scaled *y*-axis. For all box plots, statistics are based on *n* = 10 animals for each group, except for group 19, where *n* = 9. Kruskal–Wallis rank sum test performed on each dose group indicated no significant difference among the animal groups within each dose group (*p* value > 0.1). For all box plots, the median (central line) and the lower and upper quartiles (box limits) are displayed. Whiskers extend to the maximal and minimal value or, if exceeded, to max. the 1.5 × inter-quartile range. Black points represent individual animal values of one sample per animal. Source data are provided as a Source Data file.

animal and their ancestors (see "Methods"). We first differentiated the effect of treatment in the F0 generation (FDR < 0.1) of at least moderately expressed genes (estimated base mean expression > 50). This resulted in 1523 genes correlated with F0 treatment history (Supplementary Data 2). GO-term enrichment analysis and closer investigation of the underlying genes suggest a perturbation of genes related to ribosome function, chromatin modifications (including a broad range of histone demethylases and DNA methylation regulating genes such as *Dnmt3b* and *Tet* family genes), as well as tissue development and differentiation pathways (including Wnt/β-catenin, TGF-β and Notch1 signaling pathways) (Supplementary Fig. 5).

In the original Zeybel et al. paper, the reported adaptive effects on fibrogenesis were apparent in F1 and displayed cumulative F2 effects with successive F0 and F1 exposure. We thus expanded gene expression analysis to consider potentially additive F0 + F1 effects on modulating gene expression in F2. We focused on genes showing the same log2 fold-change direction for the estimated F1 treatment effect, resulting in 1304 genes with additive F0 and F1 treatment effects. We

further assumed cumulative transcriptional effects should be most apparent when comparing the control groups of cohort A vs D, and to a lesser extend of cohort A vs C (as underlined by the observations based on the PCA in Fig. 5a), narrowing down the selection to 69 genes that show FDR < 0.1 in the group 11 vs group 20 contrast and FDR < 0.2 in the group 11 vs group 17 contrast (Supplementary Data 2). While all generational effects may not be cumulative[27], this analysis complements and restricts the gene signature for more in-depth analyses and experimental validation.

We next investigated the selected 69 genes for functional enrichment and cell-type specificity. GO-term over-representation analysis identified weak enrichment for genes associated with protein deacetylation, cell-substrate junction, cell-matrix and focal adhesion, hemopoiesis, as well as regulation of fat cell differentiation (Fig. 5d). Interestingly, among the 69 genes, we find a subset of genes, such as *Ahr, Il17ra, Vegfa* or *Ski,* that are reported to play a role in hepatic stellate cells (HSCs) activation and regulation of liver fibrogenesis[28–32]. Consistent with the enrichment for genes involved

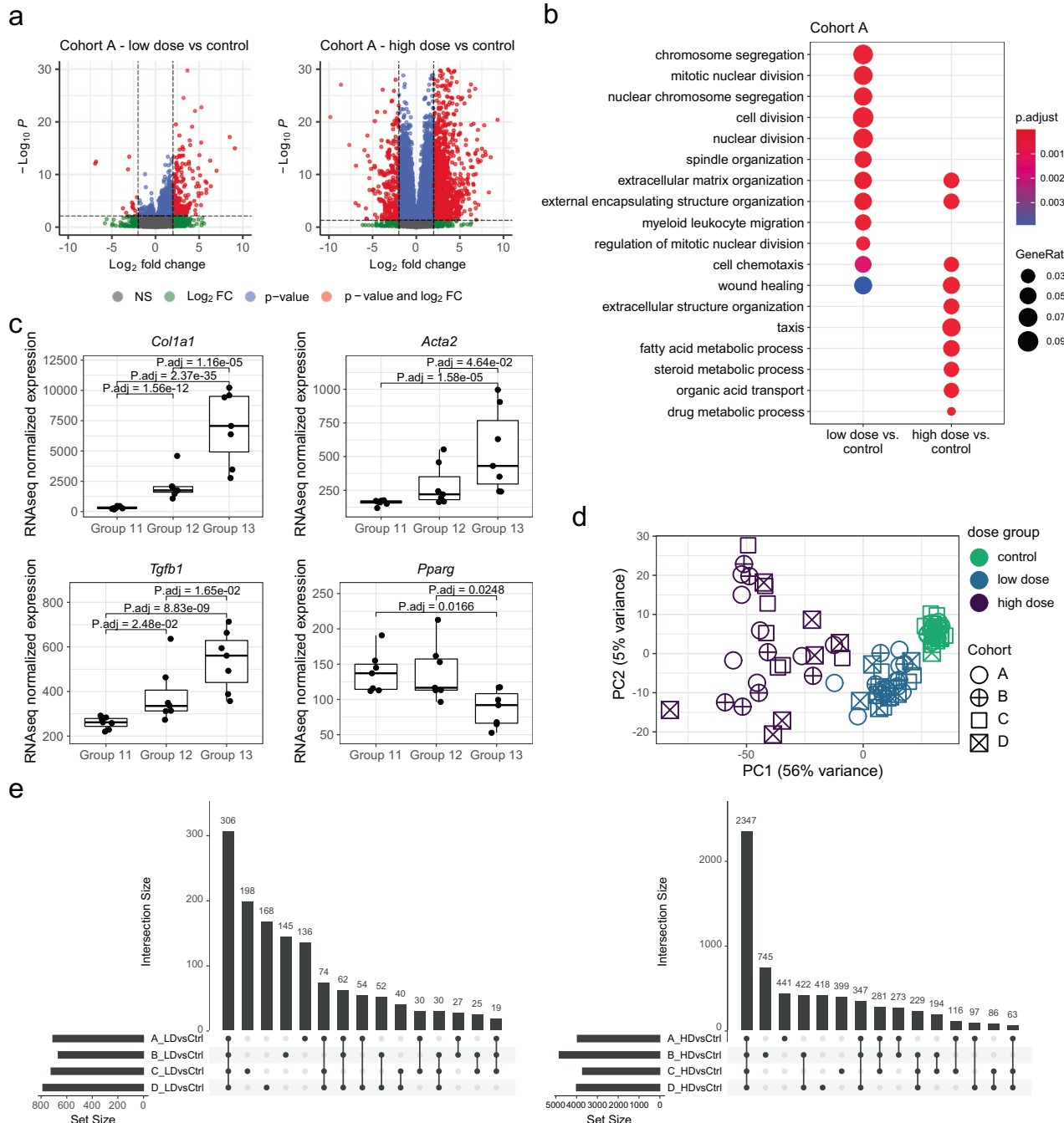

**Fig. 4 | CCl$_4$ treatment leads to extensive and dose-responsive transcriptional changes in rat liver. a–c** RNAseq analysis result illustration for cohort A. Cross-cohorts analyses are available in Supplementary Fig. 4. **a** Volcano plots illustrating differential expression analysis statistics based on DESeq2 analysis (detailed in "Methods") for low-dose (left) and high-dose (right) treatment vs control. For each gene, a point indicates the estimated log2 fold-change (treatment over control group; x-axis) and the associated negative log10 p value (y-axis). Genes are colored based on categorization upon reaching the cutoffs adjusted p value < 0.1 and/or absolute log2 fold-change (Log2 FC) > 1. **b** Comparison of one-sided GO-term over-representation test results (see "Methods") between the two dose contrasts. The dots size (GeneRatio) illustrates the ratio between the number of differentially expressed genes (contrast indicated on x-axis) and the number of GO-term-annotated genes. The color scale indicates the FDR (p.adjust) resulting from the overrepresentation test. **c** Visualization of gene expression (y-axis, DESeq2 size-factor normalized RNAseq counts) for each animal (points) by treatment group (x-axis, n = 7 animals per group) of selected liver fibrosis marker genes *Col1A1, Acta2,*

*Tgfb1*, and *Pparg*. Adjusted p value from DESeq2 analysis are indicated for selected comparisons. For all box plots, the median (central line) and the lower and upper quartiles (box limits) are displayed. Whiskers extend to the maximal and minimal value or, if exceeded, to max. the 1.5 × inter-quartile range. Black points represent individual animal values of one sample per animal. **d** Illustration of sample location on Principal Component 1 (PC1, x-axis) and PC2 (y-axis) resulting from a PCA based on the top 3000 most variable genes. For each sample (point), shape and color indicate the animal's cohort (A–D) and the F2 dose group, respectively.
**e** Illustration of overlaps in differentially expressed genes between cohorts A–D for control vs low-dose (LD, left) and vs high-dose (HD, right) contrasts. The horizontal bar plot indicates the total number of differentially expressed genes (Set size, x-axis) for each contrast. Sets/cohorts included in individual comparisons are indicated as black points connected by lines in the lower panel, while the number of genes in the intersection of the included sets/cohorts is indicated in the bar plot above (Intersection Size). Source data are provided as a Source Data file.

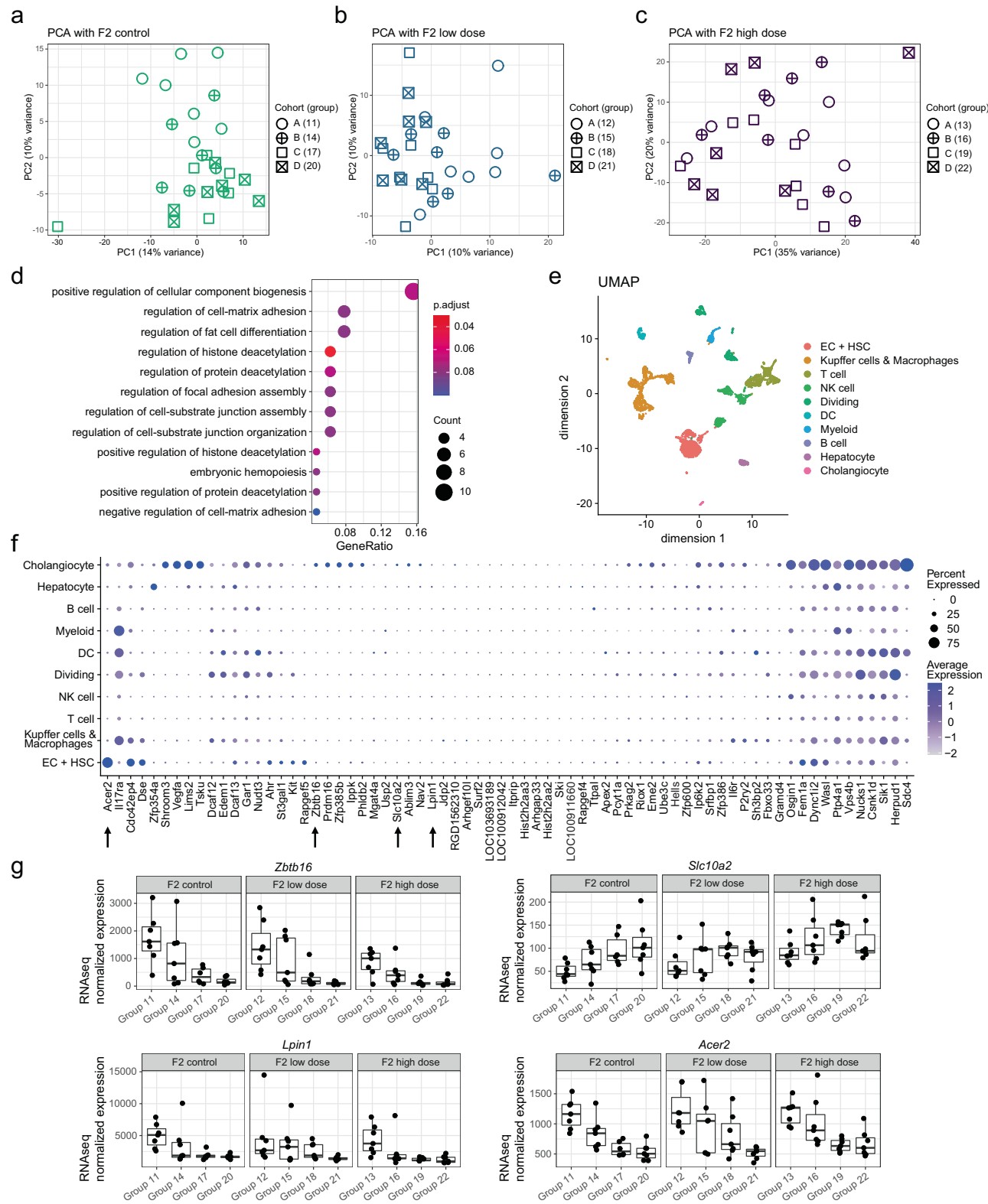

in chromatin regulation among the 1523 genes correlated with F0 treatment history, we find *Hells* (also known as *Lsh* or *Smarca6*), a member of the SNF2 DNA helicase family and important cell proliferation regulator involved in chromatin modifications and structuration[33–35] among the upregulated genes (Supplementary Fig. 6a). On the other hand, consistent with the RT-qPCR evaluation of various fibrogenic genes (Fig. 2c), we did not detect any ancestral influence on the F2 RNAseq based expression of classical fibrosis markers such as *Col1a1*, *Acta2*, and *Tgfb1* or *Pparg* expression (Supplementary Fig. 6b).

Next, we evaluated the cellular context of the 69 selected genes mapping their expression to single-cell level using publicly available rat liver single-cell RNA sequencing (scRNAseq) data[36]. The data set consisted of two pooled liver samples from three 5-month-old or two 27-month-old male naive rats fed *ad libitum*, amounting to 6045 single cells after sequencing and pre-processing. We clustered and annotated

**Fig. 5 | Identification of a transgenerational transcriptional signature of ancestral liver injury in F2 generation male offspring. a–c** Illustration of sample location on Principal Component 1 (PC1, *x*-axis) and PC2 (*y*-axis) resulting from three separate PCA based on F2 treatment conditions: control (**a**, green), low-dose (**b**, blue) and high-dose (**c**, purple). The PCA is based on top 1000 most variable genes. **d**, Illustration of the one-sided GO-term over-representation test results (see "Methods") based on the 69 genes signature (F0 + F1 cumulative effects). The *x*-axis indicates the fraction of GO-term-annotated genes that overlap with the gene set. For each over-represented GO-term, the dots size (Count) indicates the number of genes from the gene set that overlap with the GO-term-annotated genes. The color scale indicates the FDR (p.adjust) resulting from the over-representation test. **e**, Projection of the male rat liver scRNAseq data onto two dimensions using

UMAP[36]. Each cell is indicated as a point colored by its cluster annotation. Clusters were manually mapped to cell types based on expression of marker genes (see "Methods") **f** Representation of the average expression (color intensity) and percent of cells (point size) expressing the 69 genes (*x*-axis) per scRNAseq-derived cell type/cluster (*y*-axis). **g** For exemplary genes (*Zbtb16, Lpin1, Acer2, Slc10a2* arrowed in **f**), visualization of gene expression (*y*-axis, size-factor normalized RNAseq counts) for each sample (points) by treatment group across cohorts A–D. For all box plots, the median (central line) and the lower and upper quartiles (box limits) are displayed. Whiskers extend to the maximal and minimal value or, if exceeded, to max. the 1.5 × inter-quartile range. Black points represent individual animal values of one sample per animal. Source data are provided as a Source Data file.

the cells based on their overall expression pattern and known marker genes (see "Methods"), resulting in 10 distinguishable cell types including immune cell types, Kupffer cells, macrophages, endothelial cells (EC), hepatic stellate cells (HSC), hepatocytes and cholangiocytes (Fig. 5e). We next plotted the average expression and percent of expressing cells in each cluster for all genes selected in our transgenerational study (Fig. 5f). Most genes show a cell-type specific pattern rather than ubiquitous expression. Distinct subsets of the selected genes fall onto endothelial cells and HSCs, cholangiocytes, or dendritic cells (DCs), Kupffer cells, macrophages, and dividing cells. Few of our selected genes are expressed specifically in hepatocytes, myeloid cells or lymphocytes, suggesting specificity of the selected genes to cell types involved in hepatic response to injury. Global GO-term enrichment from both F0 and F0 + F1 effects, single-cell level data distribution and individual gene assessment thus collectively support potential biological relevance of the ancestral liver damage-correlating F2 liver transcriptome changes.

The ability of an environmental factor to influence the transgenerational epigenetic programming of an organ's transcriptome is precedented[37–41] and investigations of different tissue transcriptomes in male and female F3 generation vinclozolin versus control lineage rats demonstrated that all tissues had transgenerational transcriptome effects[37]. We thus evaluated whether transgenerational molecular effects could take place in other tissues through profiling the kidney transcriptome from all four ancestral vehicle groups 11, 14, 17, 20, selecting the same subset of 7 animals evaluated for liver. In a PCA, the percent of variance in gene expression profiles captured by each PC is low (≤ 23%). Excluding variance likely capturing effects resulting from sample processing (PC1 and PC2), we identify gene signatures in PC3 and PC4, albeit capturing only 10% and 5% of variance in the data, that separates group 11 (cohort A) animals (Supplementary Fig. 7). Next, we differentiated the effect of treatment in the F0 generation (FDR < 0.1) of at least moderately expressed genes (estimated base mean expression > 50) and identify 66 genes (Supplementary Data 2) whose expression is correlated with F0 history. Interestingly, 20 of those genes overlap with the F0 liver gene signature (*n* = 1523 gene set). Within those we notably find *Zbtb16* to show the same expression distribution profile as in liver (Fig. 5g). With this additional data we cannot exclude that other tissues beyond liver may be transcriptionally affected.

**Orthogonal validation of adaptive gene expression changes correlating with ancestral liver damage**

RNAseq may be subject to technical biases including RNA extraction and purification, library construction, sequencing, and bioinformatic analysis[42]. To mitigate this, we next evaluated a few exemplary genes for ancestral influence on gene expression levels in an independent and orthogonal manner. We selected genes with cumulative F0 + F1 effects and strongest estimated fold-change between groups 11 (cohort A) and 17 (cohort C), which best illustrate potential transgenerational adaptation effects from F0 to F2 in absence of F1 treatment (Fig. 1a). We selected three downregulated genes (*Zbtb16, Acer2, Lpin1*)

and one upregulated (*Slc10a2*) gene for orthogonal evaluation (Fig. 5g). *Zbtb16* (also known as *Plzf*) is a zinc finger protein acting as transcriptional repressor through its interactions with histone deacetylases[43,44] and is involved in hepatic glucose homeostasis regulation[45]. In our analyses, *Zbtb16* contributes to the GO-term enrichment for regulation of fat cell differentiation and its mapping to single-cell data suggests relevance to cholangiocytes. *Slc10a2* (also known as *Asbt*) belongs to the sodium/bile acid cotransporter family, its inhibition attenuates cholestatic liver and bile duct injury by reducing biliary bile acid concentrations in mice[46]. *Slc10a2* expression in the scRNAseq data set is also mostly restricted to cholangiocytes. *Lpin1* acts as a coactivator of PGC-1alpha/PPARα-mediated hepatic lipid metabolism[47] and influences hepatic lipid metabolism through mRNA splicing, as well as through enzymatic and transcriptional activities[48] and plays a role in the regulation of fibrogenesis and TGF-β signaling in HSCs[29]. *Lpin1* contributes to the GO-term enrichment for histone deacetylation and fat cell differentiation. Mapping to scRNAseq data indicates relevance of this gene to the hepatocyte cell population. Finally, *Acer2* is reported to be part of the p53-dependent pathway and promotes autophagy and apoptosis in response to DNA damage in adipose tissue[49]. *Acer2* contributes to the GO-term enrichment for cell-matrix adhesion and shows strong expression in the endothelial and HSC population in the scRNAseq data set.

For validation of the ancestral influence on gene expression levels of these four exemplary genes, de novo RNA extraction was performed from all frozen liver samples from groups 11 (*n* = 10) and 17 (*n* = 9, 17005 excluded). In addition, independent RNA extractions were performed from matching liver OCT samples, and cDNA prepared for quantitative RT-qPCR evaluation. The RT-qPCR evaluation from both sets of frozen and OCT samples showed gene expression changes consistent with the RNAseq, with increased *Slc10a2* and decreased levels of *Zbtb16*, *Lpin1* and *Acer2* (Fig. 6a–d). Additionally, matching time-fixed FFPE liver samples from groups 11 and 17 were evaluated by in situ hybridization (ISH) using well validated cDNA probes for the four selected genes. *Zbtb16* was found broadly distributed by ISH, including in hepatocytes, endothelial cells and cholangiocytes. Similar qualitative distribution was observed in groups 11 and 17. Quantitative image analysis results showed a trend of decreased stained surface in group 17 compared to 11 (Fig. 6a). *Slc10a2* was found expressed specifically in cholangiocytes of any size bile ducts. Some hepatocytes were also found positive in portal area albeit with low level of expression (Fig. 6b). Quantitative image analysis results for *Slc10a2* in the bile duct area showed important inter-individual variation of staining intensity within the same group (e.g., animal 11003 in group 11). The overall results showed nevertheless a trend of increased stained surface in group 17 compared to 11. *Lpin1* was expressed in all hepatocytes from both groups 11 and 17. We noted a tendency for stronger expression in portal area compared to central vein area. Quantitative image analysis results for *Lpin1* in hepatocytes area (Fig. 6c), showed strong inter-individual variations of staining intensity within both groups. The overall results showed a weak trend of decreased stained surface in group 17 compared to 11. Finally, *Acer2*

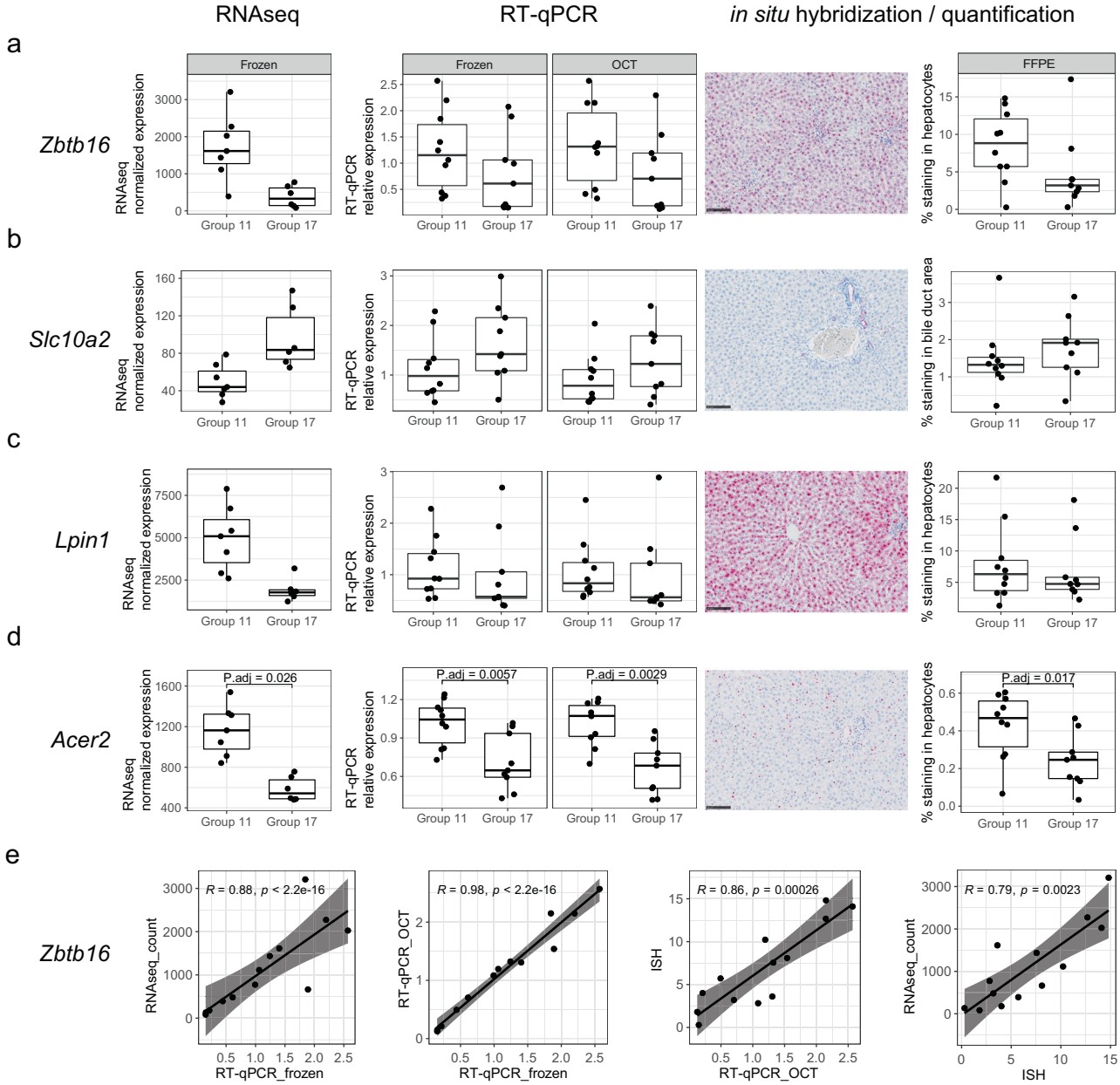

**Fig. 6 | Orthogonal evaluation of a subset of genes whose expression in F2 male offspring correlates with ancestral liver injury.** Four genes, *Zbtb16* (**a**), *Slc10a2* (**b**), *Lpin1* (**c**) and *Acer2* (**d**) were evaluated head-to-head using independent approaches and sample matrices from the study groups 11 (F2 vehicle treated, cohort A, $n = 10$ animals) and 17 (F2 vehicle treated, cohort C, $n = 9$ animals). For each gene, data from four gene expression evaluation approaches are provided: 1. RNAseq from the original RNA extraction (frozen samples). For each animal, gene expression is provided as DESeq2 size-factor normalized RNAseq counts (y-axis) for each sample (points) by treatment group (x-axis). Adjusted *p* value from the DESeq2 differential gene expression analysis (see "Methods") between group 11 and 17 are provided for cases with adjusted *p* value < 0.05, otherwise FDR ≥ 0.05; 2. RT-qPCR from a de novo, randomized, RNA extraction from frozen samples and 3. RT-qPCR from new, randomized, RNA extraction from matching OCT samples. The y-axis shows the relative mRNA expression calculated by ΔΔCT with *Hprt1* as control gene and normalized to mean of cohort A control group 11 for each animal; 4. In situ hybridization (ISH) evaluation and quantification from matching FFPE samples (y-axis represents % staining in hepatocytes for *Zbtb16*, *Lpin* and *Acer2* and % staining

in bile duct area for *Slc10a2*). Representative images (×20 magnification, scale bar 100 µm) were acquired for in situ hybridizations for each of the four genes. For RT-qPCR and ISH, exact *p* values resulting from a two-sided Wilcoxon rank sum test are indicated for significant comparisons (*p* value < 0.05), otherwise the *p* value was ≥ 0.05. For all box plots, the median (central line) and the lower and upper quartiles (box limits) are displayed. Whiskers extend to the maximal and minimal value or, if exceeded, to max. the 1.5 × inter-quartile range. Black points represent individual animal values of one sample per animal. **e**, Correlation plots evaluating cross-sample correlations across orthogonal approaches for *Zbtb16* evaluation. For each animal (points) values on x- and y-axis show measurements as indicated in the axis labels. The black line visualizes the linear regression line and the shaded areas the 95% confidence interval regions. For each comparison, Spearman's correlation coefficient (R) and significance level (*p*) of association calculated using two-sided Spearman's correlation test are provided. Full cross-readout correlation analyses for all four genes are available in Supplementary Fig. 8. Source data are provided as a Source Data file.

(Fig. 6d) was mainly expressed in isolated single cells located in the Disse space and compatible with HSCs. Some positive cells were also detected in blood vessel (endothelial cells). While qualitatively similar distribution was observed between the two groups, quantitative image analysis of *Acer2* expression in hepatocytes area showed significant decrease of the stained surface in group 17 compared to 11. Correlation analyses of individual data from all four orthogonal gene expression evaluation readouts (RNAseq on frozen samples, RT-qPCR on newly extracted frozen samples, RT-qPCR on OCT samples, and ISH on FFPE samples) showed strong inter-sample and cross-assay correlation levels supporting the robustness of the detected changes (Fig. 6e, Supplementary Fig. 8). In summary, we have identified and orthogonally validated a set of genes of potential biological relevance to cellular memory, liver homeostasis, injury and adaptation, whose expression pattern correlates with ancestral liver injury. While we could not detect any apparent morphological reduction of the liver fibrosis response and thus lack a phenotypic correlate for this adapted gene signature, we conclude that ancestral liver damage may trigger a heritable predisposition or adaptation to baseline molecular pathways in liver and potentially other tissues through germline transmission.

## Discussion

Little is known about the heritability of hepatic fibrosis. A prospective twin study had highlighted that hepatic steatosis and the level of liver fibrosis correlated between monozygotic twins but not between dizygotic twins, providing evidence that hepatic steatosis and hepatic fibrosis are heritable traits[50,51]. Such data and further characterization of functional heritable pathways that affect hepatic fibrosis have widespread implications for developing targeted genetic as well as epigenetic therapeutic approaches for the treatment or prevention of NASH-related fibrosis.

In this work, we have independently evaluated a model of transgenerational inheritance of the hepatic wound healing response[18]. In contrast to the original study, detailed pathological evaluations of liver fibrosis in F2 animals with ancestral exposure to CCl4-induced fibrosis, in a well-powered and -controlled study, do not support an adaptive phenotypic suppression of the hepatic wound healing response or greater fitness of animals with ancestral liver injury exposure. In the last couple of decades, numerous rodent models of multigenerational responses to various environmental factors or insults have been published. Whilst most studies are scientifically and technically sound, a few studies have been challenged and the majority still await independent confirmation[13]. Transient *in utero* exposure to endocrine disruptors such as vinclozolin was initially reported to lead to transgenerational (F1 through F4) effects on male fertility in inbred Fischer rats[1]. However, no transgenerational anti-androgenic effects were observed in two follow-up studies using outbred Wistar rats[12,52]. Furthermore, a study by an additional independent research group, using inbred Sprague Dawley rats, reported no effects on spermatogenesis in F1 or F2 males after maternal (F0) exposure to vinclozolin[53]. The reasons for these discrepancies have been extensively discussed, including the potential influence of genetic variations among strains, and differences in experimental designs, in particular the importance of exposure windows to vinclozolin during gestation[13,54]. A subsequent multigeneration study in mice further showed that whereas endocrine disruptors lead to epigenetic effects of the exposed germ cells, those are corrected in the mammalian germline by reprogramming events in the next generation[9]. In an alternative, well-documented model of intergenerational transmission, prenatal glucocorticoid over-exposure in pregnant female Wistar rats programs the offspring for adverse cardiovascular and metabolic effects[55]. While those effects are transmissible to a second generation through both male and female lines, they resolve in the third generation[55]. The mechanisms underlying glucocorticoid-programmed effects in F1 and F2 generations differ, with marked parent-of-origin effects in F2[27] and no evidence for

changes in classical epigenetic marks found in the male germline in this model[10]. Thus, the current weight of evidence for robust transgenerational effects in mammals is weak, and the mechanistic basis by which ancestral information may be passed to progeny despite germline and zygotic waves of global epigenome reprogramming is unclear, resulting in continued debate on the optimal study design framework for generating reproducible and high-quality data[17,21,54]. While we applied high standards and considered many potential confounding effects in running this study, a few study parameters might contribute to a different F2 phenotypic outcome when compared to the Zeybel et al. study. These include the slight difference in treatment duration and the choice of route of administration (4 weeks, twice weekly intraperitoneal injection in Zeybel et al.[18,56] versus 6 weeks, three times weekly oral gavage using comparable formulation in our study). This could produce a different plasma peak exposure profile and/or altered metabolite formation, which might account for treatment differences and phenotypic outcome. Additionally, whilst our overall study design, analyses and validation provide strong technical confidence in the transgenerational data, we cannot fully exclude that transmissible germline effects (e.g., secondary (epi-)mutations) might take place. The primary site of CCl4 toxicity is the liver. However, CCl4 was also reported to cause functional damages in other organs of the body[57]. Whilst no gross testis effects or significant sperm count changes were observed, CCl4 could hypothetically cause oxidative damage in reproductive cells and tissues[24] and represent a treatment-related confounding effect when evaluating the effects of ancestral liver damage. Further evaluation of the phenomenon would ideally need to consider extended treatment durations, different lengths of intergenerational recovery periods, an assessment of the potential for maternal transmission and the potential for maintenance and impact of effects on a F3 generation, although logistical and resource challenges generally preclude the feasibility of such studies.

Despite the lack of an apparent liver fibrosis adaptation or measured phenotypic correlate, we identify in F2 animals a differential transcriptional liver gene set that correlates with ancestral CCl4-induced liver injury, suggesting transgenerational transmission at the molecular level. Orthogonal evaluation of a subset of genes that best illustrate potential transgenerational adaptation effects from F0 to F2 in absence of F1 treatment (i.e., groups 11 vs 17) point to consistent hepatic expression changes, regardless of the liver sample matrix (frozen, OCT, FFPE) and readout (RNAseq, RT-qPCR, ISH). Our collective hepatic expression data (GO-term enrichment, single-cell level data distribution and individual gene assessment) point to programming of various molecular pathways of potential biological relevance to cellular memory, liver homeostasis, injury and adaptation. Interestingly, the altered expression of epigenetic (co-)effectors such as histone and DNA demethylases or *Hells/Lsh* provides the possibility that epigenetic processes could be involved. *Hells* is directly required for the methylation and silencing of transposable repetitive elements during gametogenesis and in somatic cells[58] and its mis-regulation was linked to male germline epigenetic changes following paternal irradiation, resulting in deleterious effects in the somatic thymus tissue from the progeny of exposed animals[59]. The functional relevance of the identified gene sets on liver fibrosis adaptation and the nature and transmission of potential germline and somatic (epi-)genomic changes remain unclear and require further investigations. The ability of an environmental factor to influence the transgenerational epigenetic programming of an organ's transcriptome is precedented[37-41]. In a study by Anway et al., transcriptome characterizations identified 196 genes to be differentially expressed in F1–F3 testis of animals ancestrally exposed to vinclozolin[41]. Affected genes had roles in chromatin regulation, including significant reductions of *Dnmt3a* in F1–F2, *Dnmt1* in F1–F3, *Dnmt3L* in F1–F3, and *Ehmt1* in F1–F3. Interestingly, investigation of different tissue transcriptomes in male and female F3

generation vinclozolin versus control lineage rats demonstrated that all tissues had transgenerational transcriptome effects[37]. We have further evaluated the concept that heritable transcriptional effects could occur in tissues beyond the liver, and identified minimal, but plausible transgenerational kidney transcriptional variation following CCl$_4$-induced liver fibrosis, including at *Zbtb16*. Thus, effects transmitted through the germline may affect the transcriptome of multiple tissues, although the biological significance of the apparent transcriptional transmission and uncertainty on underlying mechanisms limit definitive conclusions.

The possibility that ancestral environmental conditions and exposures can result in effects that are inherited across generations has led to significant public awareness and regulatory attention, including debates on both ethical and legal implications[60,61]. Transgenerational effects represent an important theoretical safety assessment consideration for emerging therapeutic modulators of epigenetic effectors, as highlighted by recent publications assessing the mutigenerational impact of valproic acid[62,63] and emphasized by a recent pharma industry epigenetic working group survey[64] facilitated by IQ DruSafe. However, the toxicological relevance of transgenerational effects for therapeutic drugs remains uncertain and robust paradigms will be necessary to evaluate this phenomenon. Our re-evaluation of intergenerational effects of CCl$_4$ exposure in rats provide evidence for transgenerational transmission of ancestral liver injury at the molecular level but no heritable morphological phenotypic effects on the liver wound healing were observed thus highlighting the need for further evaluation of transgenerational epigenetic inheritance paradigms in mammals.

## Methods

### Ethical statement
The study has been conducted in accordance with the Novartis Animal Care and Use Committee. All procedures in this study are in compliance with the Animal Welfare Act, the Guide for the Care and Use of Laboratory Animals, and the Office of Laboratory Animal Welfare. Key elements of the study protocol are detailed in Supplementary Methods.

### Animals
Animals used in this study were outbred Sprague Dawley rats supplied by Charles River Laboratories Raleigh, NC. Ordered F0 (males, females) and F1 (females) were supplied by Charles River Laboratories Raleigh, NC approximately 10 weeks of age (Supplementary Methods, Tables 1–4). Throughout the study males were treated at 9-11 weeks of age. Detailed study schedules for all three generations are provided in Supplementary Methods (F0: Table 1–8, F1: Table 1–9, F2: Table 1–10). For the full study period, 2 or 3 animals of the same sex were co-housed per cage except during mating and lactational periods. Females were individually housed with their litters during lactational periods. During gestation, females were housed 2–3 per cage. Animals were single housed on gestation day 20 in preparation for parturition.

### Mating and animal allocation
Timed mating was initiated by cohousing 1 female and 1 male for up to 2 weeks. Previously treated males were allowed to recover for 14 days before the initiation of mating. Females were checked daily for confirmation of mating and mating was considered to have occurred when sperm was observed in the vaginal washing and/or a vaginal plug was observed. Once mating had occurred, females were returned to their home cages. On day 4 post-partum all F0 and F1 litters were randomly culled to 8 pups/litter with male pups preferentially retained. Two male pups/F0 litter were selected to comprise the F1 generation and three male pups/F1 litter were selected to comprise the F2 generation. If this was not possible due to the number of pups, a maximum of 2 pups from a litter were assigned to a dose group.

### Experimental model for CCl$_4$-induced liver fibrosis treatment
Male animals were dosed three times weekly (t.i.w) for 6 weeks via oral gavage as per established and reproducible CCl$_4$-induced internal protocol. The high-dose groups received 0.5 mL/kg 50% (v/v) CCl$_4$ in olive oil. The low-dose groups received 0.5 mL/kg 8% (v/v) CCl$_4$ in olive oil. The control groups received 0.5 mL/kg 100% olive oil. Sample collection from F0 and F1 animals was performed after recovery and successful mating, 4 weeks following the last dose. Sample collection from F2 animals was done 24 hours (h) following the last dose at a time point referred to as peak fibrosis in the original study[18].

### Clinical observations
Animals were checked for mortality and moribundity at least once daily. In all generations, clinical observations were made once daily on non-dosing days and at least twice daily on dosing days. The body weight was measured twice weekly. Food consumption was calculated twice weekly. One animal in the F0 generation (CCl$_4$ treated animal 2005) and two animals in the F2 generation (vehicle control treated animal 17005 and CCl$_4$ treated animal 19003) did not tolerate the treatment and were excluded from the analyses. Detailed in Supplementary Methods.

### Sperm count
The left testes of all animals were collected, decapsulated, weighed and frozen at necropsy for sperm count evaluations. For sperm count, testis was homogenized in 10% Triton X-100 in saline solution. The homogenate was diluted to 50 mL in saline and sperm heads were counted with a hemocytometer. Two counts were performed for each sample and the average was used to calculate the sperm per gram testis as follows:

[(average number of sperm heads counted) * (squares factor: 5) * (hemocytometer factor: 10$^4$) * (dilution factor: 50)/(tissue weight of testis in grams)]

### Clinical pathology/biochemistry
Blood samples for clinical pathology were taken one day after last dose of the t.i.w. dosing scheme for all animals. For F0 and F1 additional samples were taken at the end of the two weeks recovery period (day 53) and at necropsy post-mating on day 74. Detailed in Supplementary Methods.

### Anatomic pathology
At the completion of the study, all animals were submitted for necropsy and kidney, liver, and skin were collected and put in formalin jar after opening the abdominal cavity. The right testes of all animals were collected and put in modified Davidson's fluid. Detailed in Supplementary Methods.

### H&E staining and analysis
Tissues were fixed in 10% buffered formalin for 48 h at room temperature and processed for embedding in paraffin using standard procedures. Sections were stained with hematoxylin and eosin (H&E) using standard protocols. Digital images were obtained with a ScanScope XT system (Leica, Nussloch, Germany).

### Sirius Red staining and analysis
Tissues were fixed in 10% buffered formalin for 48 h at room temperature and processed for embedding in paraffin using standard procedures. For Sirius Red staining, Picrosirius Red solution (EMS, Hatfield, PA, USA) was used according to the manufacturer's recommendations. Stained tissue sections were captured as whole slide scans using a Leica AperioAT2 slide scanner using 40x magnification with resulting image pixel dimensions of 0.25 μm (x, y).

## DenseNet analysis

Quantitative assessment was performed using HALO and HALO AI software (v.2.3) in two steps: pixel classification of the tissue regions of interest at low resolution (random forest classifier) and subsequent pixel classification of sub-structures at higher resolution (DenseNet classifier). As control and treated groups exhibited vastly different morphological features, two classifiers were trained individually. Final readouts given as percentages reflect area ratios.

## RNA sequencing

RNAseq experiments were performed by Genesupport/FASTERIS SA, Plan-les-Ouates, Switzerland. Total RNA extraction from 50 to 100 mg tissues was performed by tissue grinding in liquid nitrogen, homogenization by bead beating, proteinase K digestion and purified using the Qiagen RNeasy kit. Sequencing was done using a PolyA RNA sequencing protocol with 50 bp paired-end Illumina reads. Next generation sequencing libraries were prepared with the TruSeq Stranded mRNA Sample Preparation kit (Illumina) from 500 ng of input RNA according to the manufacturer's instructions, using 12 (liver) or 13 (kidney) cycles of amplification. The resulting libraries were pooled and loaded on an NovaSeq 6000 (Illumina) for paired-end 50 bp sequencing, generating 47 million reads per sample on average (liver) and 37 million reads per sample on average (kidney).

## RNAseq analysis

Salmon (version 0.14.0) was used to quantify gene expression from fastq files. The index was generated from *Rattus norvegicus* Ensembl release 100 with parameter −k 31. Each sample was quantified independently, providing fastq files from all sequencing runs if present for the sample, and the following parameters: -l ISR --validateMappings --gcBias. The quantification results were loaded in R (version 3.5.0) using tximport (1.10.1) to summarize quantification results by gene. All exploratory and differential gene expression analysis was performed in R (version 4.1.1) using DESeq2 (version 1.32.0)[65]. Only genes showing more than 10 reads in at least one sample group across the full liver (groups 11–22) or kidney (groups 11, 14, 17, 20) data sets were considered for analysis. For PCA, gene counts were transformed with variance stabilization on the full liver or kidney data set. For individual analyses, the transformed data was sub-selected to the samples relevant for the explorative analysis. The design for de-convoluting the F0 and F1 effect in the differential expression analysis with DESeq2 was set to ~ F0 + F1 + F2, where F0 and F1 variables indicate treatment or control animals and F2 indicates control, low-dose, or high-dose animals. For identifying F2 treatment effects, the design was set to ~ group, where the group variable indicates for each animal its treatment/cohort group combination (groups 11–22). Results were plotted in a volcano plot using the package EnhancedVolcano (version 1.10.0). Overlaps between differentially expressed genes between contrasts were evaluated using the UpSetR package (version 1.4.0). GO-term over-representation analysis was performed using the clusterProfiler package (version 4.0.2) and the Biological Pathway sub-ontology (org.Rn.eg.db version 3.13.0) in the enrichGO function. Redundant pathways were simplified using clusterProfilers' simplify function with a cutoff of 0.7 for similarity and only the GO-term with minimal FDR is reported in the results. The Rmarkdown containing all analysis code is available on https://github.com/jperner/TGmanuscript (https://doi.org/10.5281/zenodo.8321732).

## Single-cell RNAseq analysis

Pre-processed matrix, barcode, and gene files for samples GSM4331834 (young male) and GSM4331835 (old male) were downloaded from NCBI's Gene Expression Omnibus (GEO) (see "Data availability" Statement). The R (version 4.2.0) package Seurat (version 4.1.1) was used for processing the data. Only cells with nFeature_RNA > 500 and less than 15% mtRNA content, as well as nCount_RNA < 10 * nFeature_RNA were retained. The data were log-normalized and scaled. The top 2000 most variable features were selected after applying variance stabilization. The top 45 most important principal components were selected based on a JackStrawPlot and passed to the FindNeighbors and FindClusters function (setting resolution = 0.1) for cell cluster identification. Uniform Manifold Approximation and Projection (UMAP) was run with default parameters to visualize the resulting clustering in a two-dimensional space. We have used markers described in several publications to manually annotate cell types[36,66–69].

## RT-qPCR

RNA was isolated from snap-frozen tissue and OCT embedded samples using Qiazol followed by RNeasy Mini or RNeasy Micro kit (Qiagen). Samples were DNase I treated, concentration was determined by Qubit measurement and quality was assessed using Agilent Bioanalyzer. RNA isolation was randomized for snap-frozen tissue including animals of all F2 groups in each of the four RNA isolation batches. RNA isolation for OCT samples was done in one batch. cDNA was generated from 500 ng of RNA by using High-Capacity cDNA Reverse Transcription Kit (ThermoFisher) following the manufacturer's instructions. For RT-qPCR, 6 ng of cDNA were used per reaction. The following Taqman assays were used: *Acer2* (Rn01770005_m1), *Acta2* (Rn01759928_g1), *Col1a1* (Rn01463848_m1), *Col1a2* (Rn01526721_m1), *Hprt1* (Rn01527840_m1), *Lox* (Rn01491829_m1), *Lpin1* (Rn01469024_m1), *Slc10a2* (Rn00691576_m1), *Timp1* (Rn01430873_g1), *Zbtb16* (Rn01418644_m1). Relative expression was calculated by ΔΔCT with *Hprt1* as control gene and normalized to snap-frozen cohort A group 11 or group 12 as control group as stated per figure.

## Immunohistochemistry (IHC)

For the αSMA immunohistochemistry (IHC) staining, the 3 μm thickness liver tissue sections were deparaffined, rehydrated and stained using a Ventana Discovery® ULTRA instrument (Roche Diagnostics). An antigen retrieval pretreatment was performed using Cell Conditioning Solution CC1 (Roche Diagnostics, 06414575001) for 32 min. Non-specific antibody binding was blocked with a casein solution for 32 min. Subsequently, the mouse monoclonal primary anti- αSMA antibody (Dako, M0851, clone 1A4, batch 41327852) was applied at 1/4000 (approx. 20 ng/mL) for 60 min at 37 °C. The specificity of the antibody was tested by the provider using SDS-PAGE immunoblotting of an alpha-smooth muscle actin (SMA) and detection of the corresponding band. For usage on rat tissue, a BLAST sequence alignment found 100% sequence homology and staining results on rat liver are consistent with smooth muscle cells staining. The signal was detected with a polymer-based secondary antibody OmniMap anti-Mouse HRP (Roche Diagnostics, 05269652001) as ready to use reagent applied following the provider recommendation and ChromoMap DAB kit (Roche Diagnostics, 05266645001). The fibrosis detected in the αSMA stain was scored visually on a scale from 1 to 4. Animal 20003 was excluded from the analysis for technical reasons. The αSMA protein detected by IHC is encoded by the *Acta2* gene and is predominantly expressed in smooth muscle but is also found in myofibroblasts.

## In situ hybridization (ISH)

For ISH, the formalin fixed paraffin embedded (FFPE) blocks of F2 liver were sectioned at 3 μm and collected on SuperFrost® Plus slides. The genes *Lpin1* (probe 1124489), *Zbtb16* (probe 508899), *Slc10a2* (probe 1144679) and *Acer2* (probe 856849) were investigated on mRNA level using RNAscope® ISH technology commercialized by Advanced Cell Diagnostics Ltd. FFPE tissue sections were placed in a Discovery® Ultra instrument (Roche Diagnostics Schweiz AG, Rotkreuz, Switzerland) and processed using the mRNA Universal procedure with predefined parameters for deparaffinization, demasking, hybridization and amplification steps (with an amplification 5 step set at 2 h). Slides were counterstained with Hematoxylin II and Bluing reagent for 8 min each,

dried at 60 °C, dipped briefly in pure xylene and mounted using Eco-Mount medium. Stained slides were scanned for subsequent image analysis.

## Statistics and reproducibility
In the design of the in vivo study, the animal group sizes were not predetermined using any specific statistical method. We however considered important study reproducibility features such as animal house and care, high study power and F0–F2 pedigree tracing. Randomization was applied at various stages of the study. Given the large number of F2 animals ($n = 120$) and to avoid treatment or collection biases, we used a carefully designed staggering and randomization scheme for F2 dose–response treatment, necropsy, collection and evaluation, including for molecular profiling. The investigators were not blinded to group allocation during data collection and analysis. Knowledge of the F0–F1 liver fibrosis history represents the core anchor to interpreting pathology and molecular effects and evaluating the transgenerational phenomenon. Three animals (F0 animal 2005, F2 animals 17005 and 19003) did not tolerate the treatment and were excluded from the analyses, animal 20003 was excluded from IHC analysis for technical reasons. Detailed description of study design and animal-level information is provided in Supplementary Methods. Statistical analyses were performed in R (version 4.1.1). Detailed description of experimental and statistical tests are available in (supplementary) figure legends and "Methods" sections.

## Reporting summary
Further information on research design is available in the Nature Portfolio Reporting Summary linked to this article.

## Data availability
RNA sequencing data originating from this study have been deposited in NCBI GEO under the accession code: GSE229524. Single-cell RNA sequencing data analyzed in this study was accessed from NCBI GEO under the accession code: GSE137869 and the sample sets GSM4331834 (young male) and GSM4331835 (old male) were selected and processed from count matrix as stated in the "Methods" section. Source data are provided with this paper.

## Code availability
Custom analysis scripts generated within the course of this work are publicly available and deposited under: https://github.com/jperner/TGmanuscript and are stored permanently with Zenodo (https://doi.org/10.5281/zenodo.8321732).

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

## Acknowledgements

The authors would like to thank Ulrike Naumann, Kayhangabriel Akyel, Karine Bigot and Johann Mueller for samples and data handling and management. Additionally, we would like to acknowledge Marilynn Schneider, Keith Di Petrillo, Robert Arch and Daniel Lapadula for valuable scientific input and for project administration and planning. This study was sponsored by Novartis, Biomedical Research, USA and Switzerland.

## Author contributions

All authors contributed to the research, analysis, interpretation and writing of the final manuscript. R.T., J.B., A. Piaia., A.W.N. and L.M., designed the multigenerational study, developed the research protocol, sampling and evaluation scheme; R.T., J.B., J.P. and L.P. coordinated the overall study analytical scheme and integrated all the data. A.Piaia., A.D., A.W.N., performed, analyzed, and interpreted the clinical pathology and pathology data, including quantitative image analyses; J.B. and L.P. supported the transcriptomic experiments. J.B. and J.P. analyzed RNA-seq data and interpreted results. M.A.G., A. Piequet., V.D., performed molecular evaluations and analysis (RT-qPCR, IHC and ISH). R.T., J.M., S.D.C, developed the idea and supervised the project.

## Competing interests

The authors declare no competing interests. All authors were employees of Novartis Pharma AG during the course of this work, as indicated in the affiliations.
