## [Peer Review File · Nature Communications]

Unaltered hepatic wound-healing response in male rats with ancestral liver injuryREVIEWER COMMENTS

Reviewer #1 (Remarks to the Author):

In mammals, germ cells transmit genetic information in the form of DNA from one generation to the next, reprogramming of the epigenome, which erases possible epigenetic memory, in primordial germ cells (PGCs) and following fertilisation represents a major barrier to would-be epigenetic inheritance. Despite global hypomethylation, evolutionarily young and potentially hazardous retroelements remain methylated and silenced, preventing genetic instability that could alter gene expression in target tissues and lead to physiological deficits.

Despite the existence of this safety net in mammals, there is ongoing robust debate over the importance of germline epigenetic effects in the non-genomic transmission of phenotypes across generations. Most of the cases reported in plants and mammals tend to be driven by rare stochastic events and are not systematic when investigated methodically.

However, for transgenerational effects, the germline should have undergone a normal round of epigenetic reprogramming in the absence of any environmental insult, suggesting that any induced effects must be preserved, maintained and transmitted to a subsequent generation in the absence of an ongoing environmental influence.

In these models, the penetrance of the phenotype is high—indeed, effects are found using very small numbers of animals—but the percentage DNA methylation changes that are reported in sperm are very low to non-existent. In a recent study published in *Genome Biology* (PMID 29636086), Cartier and colleagues investigated the adverse effects of early life exposure in rats to the synthetic glucocorticoid dexamethasone resulting in cardiovascular and metabolic systems in the offspring. These programmed effects (including reduced birth weight) are transmissible to a second generation through both male and female lines, suggesting germline transmission. Although effects on birth weight are transmitted to the F2 generation through the male line, genome wide analysis could not detect differences in DNA methylation, histone modifications (H3K4me3, H3K4me1, H3K27me3 and H3K9me3) or small RNA's populations between germ cells and sperm from Dex-exposed animals and controls. These data suggests that direct transmission of changes in DNA methylation/common histone modifications/small RNAs is unlikely to be the underlying mechanism for the transmission of the phenotype in this model. Alternatively, the transmission of the phenotype occurs in the absence of epigenetic perturbations in the exposed germline. The phenotypic transmission may be dependent on signalling pathways mediated by factors in seminal fluid, the influence of paternal behaviours on the mother, microbiome transfer or the transmission of metabolites.

The report by Zeybel and colleagues (PMID: 22941276) appears to be another example of phenotypic transmission of exposure to an adverse environmental agent, carbon tetrachloride (CCl4). They reported that a history of liver damage corresponds with transmission of an epigenetic suppressive adaptation of the fibrogenic component of wound healing to the male F1 and F2 generations. Remodelling of DNA methylation and histone acetylation at candidate genes were reported to underpin these alterations in gene expression. They suggested that a myofibroblast-secreted soluble factor stimulates heritable epigenetic signatures in sperm so that the resulting offspring better adapt to future fibrogenic hepatic insults.

The submission to NCOMMs by Beil and colleagues is a comprehensive, well-planned and high quality analysis investigating the potential transgenerational effects of CCl₄ on the hepatic wound-healing response in rats. They repeated the original multigenerational study with increased power, full pedigree tracing, and a randomization scheme and F₂ dose-response evaluation. However, they were unable to detect adaptive phenotypic suppression of the hepatic wound healing response or a greater fitness of animals with ancestral liver injury exposure. This mitigates the impact of the original report. Subsequential studies like this need the full exposure and audience bandwidth provided by high quality journals such as NCOMMs. The authors offer a glimmer of hope for the CCl₄ study by identifying a ghost imprint of 69 genes (by bioinformatics analysis of F₂ liver RNA sequencing) whose expression correlates with ancestral liver injury. However, this result is a far remove from the robust phenotypic inheritance originally reported in the Zeybel paper (PMID: 22941276).

I fully support acceptance of this important manuscript by NCOMMS if the following points are addressed.

1. In the introduction I challenge the authors suggestion that 'environmental experiences.....are written into our epigenomes and can be transmitted through the male and/or female germline to influence development and health of progeny.' Outside of imprinted genes (which are a special case not involving environmental exposure, but may involve transcriptional patterning in the oocyte that lay down methylation imprints) the evidence is poor to non-existent. In this respect the Cartier (PMID 29636086) and Igal (ref 45 in the manuscript) papers are importance references and should be fully discussed in BOTH the introduction and discussion.
2. I applaud the authors in identifying a ghost set of 69 genes (group 11 vs group 20) that are prominent after extensive bioinformatics analysis. However, it does raise the question what would happen in other tissues from group 11 vs group 20, is there transcriptional instability present there also, given that many of the targets are not specifically expressed in hepatocytes (from a very nice single cell analysis).
3. In their bioinformatic RNA seq analysis the authors should check if there are alterations in repeat expression analysis (LINE1, IAP and ERV sequences), this relates to point 4 below. See PMID: 23043114 for an example of repeat analysis in the rat genome.
4. The authors seemed to have missed a trick in not highlighting one of the factors that they identified in the ghost set, notably Lsh/Hells, which they have previously reported on being upregulated in response to Phenobarbital exposure in the liver, which should be referenced (PMID: 24690595). This is of great interest because Hells is an epigenetic co-regulator for global DNA methylation levels and is required to maintain repression of IAP LTR and satellite repeat expression in mouse somatic cells (PMID: 24367978). Its mis-regulation is also linked with epigenetic changes in the male germline after paternal irradiation, which results in deleterious effects in the somatic thymus tissue from the progeny of exposed animals (PMID: 19959559). These observations are highly relevant to their study and should be discussed fully.
5. I note that the labelling for Slc10a2 is cut off in figure 5b.
6. I think the discussion should be more robust in scope, its actually good news if TGI due to environmental exposure appears to be a minor phenomenon in animals at best. As in the CCl₄ exposure model, initial observations have not been completely validated. Here, the

Cartier and Iqbal studies should be fully discussed and highlighted, it is interesting that the phenotypic consequences of prenatal glucocorticoid overexposure is well documented (<https://pubmed.ncbi.nlm.nih.gov/?term=prenatal+glucocorticoid+overexposure&sort=pubdate>). But for the glucocorticoid rat model in the F2 generation, although birth weight is also reduced, the effects on foeto-placental growth and gene expression differed from those in the F1, with marked parent of origin effects (PMID: 22086116). This implies that the mechanisms underlying glucocorticoid programmed effects in F1 and F2 generations differ. Notably the glucocorticoid-programmed phenotype did not persist into a third generation (F3) (PMID: 15178540). There were no differences between F3 offspring groups in birth weight, litter number, or gestation length or postnatal growth patterns, which is also good news and should be discussed.

If a revised version of this manuscript is accepted by NCOMMS, it certainly deserves an accompanying news and views comment highlighting its significance.

Reviewer #2 (Remarks to the Author):

In this study, the authors performed an extremely well-controlled study of the inter-generational effects of CCL4 exposure in rats, repeating a previous study that showed heritable morphological phenotypic effects on liver fibrosis response. The morphological effects were not repeated, but gene expression effects were apparent across generations, that correlated with direct and ancestral treatments. The methods are, for the most part, appropriate, and gene expression results were validated by several methods. The overall results are sound.

This is an important paper, even though a portion of it gives 'un-sexy' negative results. The field of heritable epigenetic effects on gene expression has suffered from criticism that controls, validation and experimental design are inadequate, and this study addresses those concerns well.

I have only one concern: in Lines 269-271 the authors state "We assumed that F0, F1 and F2 effects on modulating gene expression in the F2 generation are additive, i.e., resulting in cumulative changes across generations." I do not think that this is a valid assumption, as epigenetic and gene expression changes are seen to be quite different in F1, F2 and F3 generations in previous rat studies. It might be better to not limit the evaluated dataset with this parameter.

A very interesting and important study for those interested in the emerging field of neo-Lamarckian heritability.

Eric Nilsson
Washington State University

Reviewer #3 (Remarks to the Author):

Beil et al present a carefully designed and performed multigenerational study on the effects of ancestral CCL4 injury, with increased power, full pedigree tracing, randomization scheme

and an F2 dose-response evaluation. Their data do not support adaptive phenotypic suppression of the hepatic wound-healing response or a greater fitness of animals with ancestral liver injury exposure as originally reported. The authors suggest that there is an influence of ancestral injury on hepatic gene expression, but this part of the study remains vague, statistically borderline and without clear interpretation/guidance:

Comments:

1. The transgenerational study is carefully performed with sufficiently large enough cohorts and corrects a previous study that includes only small cohorts of rats (n=5).
2. The transcriptomics studies remain vague. While the authors point out transcriptomic differences that they describe as “associated with molecular pathways underlying liver homeostasis and fibrosis”, the GO terms in Fig.4D are not directly linked to fibrosis or liver homeostasis, many of them very broad and not linked to fibrosis or homeostasis or sounding vaguely like ECM or fibrosis but not really linked to fibrosis (e.g. regulation of cell-matrix adhesion, regulation of cell junction, , focal adhesion....). Likewise, the genes shown in Fig.4 and 5 are not well known central players in homeostasis or fibrosis. Most importantly, many of the data that are shown are not statistically significant. I would suggest to use strict statistical criteria such as FDR<0.05 for RNA-seq and p-values<0.05 for single genes. If one then focuses on homeostasis-relevant genes, literally nothing remains. Unless the authors have functional studies where they can prove the relevance of these genes. To the reviewer’s eyes, this looks like a desperate attempt to rescue the transgenerational concept or to provide some data to appease authors from previous publications. The authors need to show some hard-core data what the transgenerational difference and their meaning are. This is not convincing and could be due to chance given the low significance.
3. The concept of ancestral CCL4 injury exclusively affecting the liver is not convincing. The reviewer is aware of the previous Nat Med paper, but the authors should check the concept that the transgenerational effects could be on other organs as well, as effects are transmitted through the germline rather than the hepatocyte or other liver cells. As the authors preserved additional organs, it would be interesting to run one additional RNA-seq in another organ and compare differences in gene expression. This could be simply the untreated groups 11,14,17 and 20.
4. It is understood that the authors follow some of the previous groups in their protocols, but it would have been more convincing to have longer treatments of CCl4 in F0 and F1 to allow for stronger potential transgenerational effects.
5. Not to beat a dead horse, but since this is a mostly negative study, it would be helpful to confirm the histological fibrosis in Figure 2b by one simple additional test, aSMA IHC and morphometric quantification. This works very well and will help to convince the more critical reviewers. Likewise, it would be helpful and very easy to add more fibrogenic genes including Col1a2, Acta2, Lox and TIMP1 (with existing material/cDNA, a matter of a few days to get the data).
6. Since regeneration and inflammation are an important aspect for outcomes and inevitably follow injury, I would also suggest to determine proliferation by Ki67 IHC, CD45 IHC and qPCR for a few select genes.
7. Depending on where this is heading, the authors may need to change the title. The current title is weak as it is not backed by strong findings and as it does not carry the key message of unaltered fibrosis.
8. With all above, the authors should also carefully revisit whether there is possibly a lower degree of fibrosis/inflammation/proliferation in the group 12 vs 15, 18 or 21 (which would be the opposite of what was reported in the 2012 Nat Med paper).

REVIEWER COMMENTS - Beil, Pfaller, Perner et al. NCOMMS-22-45904

Reviewer #1 (Remarks to the Author):

In mammals, germ cells transmit genetic information in the form of DNA from one generation to the next, reprogramming of the epigenome, which erases possible epigenetic memory, in primordial germ cells (PGCs) and following fertilisation represents a major barrier to would-be epigenetic inheritance. Despite global hypomethylation, evolutionarily young and potentially hazardous retroelements remain methylated and silenced, preventing genetic instability that could alter gene expression in target tissues and lead to physiological deficits.

Despite the existence of this safety net in mammals, there is ongoing robust debate over the importance of germline epigenetic effects in the non-genomic transmission of phenotypes across generations. Most of the cases reported in plants and mammals tend to be driven by rare stochastic events and are not systematic when investigated methodically.

However, for transgenerational effects, the germline should have undergone a normal round of epigenetic reprogramming in the absence of any environmental insult, suggesting that any induced effects must be preserved, maintained and transmitted to a subsequent generation in the absence of an ongoing environmental influence.

In these models, the penetrance of the phenotype is high—indeed, effects are found using very small numbers of animals—but the percentage DNA methylation changes that are reported in sperm are very low to non-existent. In a recent study published in *Genome Biology* (PMID 29636086), Cartier and colleagues investigated the adverse effects of early life exposure in rats to the synthetic glucocorticoid dexamethasone resulting in cardiovascular and metabolic systems in the offspring. These programmed effects (including reduced birth weight) are transmissible to a second generation through both male and female lines, suggesting germline transmission. Although effects on birth weight are transmitted to the F2 generation through the male line, genome wide analysis could not detect differences in DNA methylation, histone modifications (H3K4me3, H3K4me1, H3K27me3 and H3K9me3) or small RNA's populations between germ cells and sperm from Dex-exposed animals and controls. These data suggests that direct transmission of changes in DNA methylation/common histone modifications/small RNAs is unlikely to be the underlying mechanism for the transmission of the phenotype in this model. Alternatively, the transmission of the phenotype occurs in the absence of epigenetic perturbations in the exposed germline. The phenotypic transmission may be dependent on signalling pathways mediated by factors in seminal fluid, the influence of paternal behaviours on the mother, microbiome transfer or the transmission of metabolites.

The report by Zeybel and colleagues (PMID: 22941276) appears to be another example of phenotypic transmission of exposure to an adverse environmental agent, carbon tetrachloride (CCl4). They reported that a history of liver damage corresponds with transmission of an epigenetic suppressive adaptation of the fibrogenic component of wound healing to the male F1 and F2 generations. Remodelling of DNA methylation and histone acetylation at candidate genes were reported to underpin these alterations in gene expression. They suggested that a myofibroblast-secreted soluble factor stimulates heritable epigenetic signatures in sperm so that the resulting offspring better adapt to future fibrogenic hepatic insults.

The submission to NCOMMs by Beil and colleagues is a comprehensive, well-planned and high quality analysis investigating the potential transgenerational effects of CCl4 on the hepatic wound-healing

response in rats. They repeated the original multigenerational study with increased power, full pedigree tracing, and a randomization scheme and F2 dose-response evaluation. However, they were unable to detect adaptive phenotypic suppression of the hepatic wound healing response or a greater fitness of animals with ancestral liver injury exposure. This mitigates the impact of the original report. Subsequential studies like this need the full exposure and audience bandwidth provided by high quality journals such as NCOMMS. The authors offer a glimmer of hope for the CCl4 study by identifying a ghost imprint of 69 genes (by bioinformatics analysis of F2 liver RNA sequencing) whose expression correlates with ancestral liver injury. However, this result is a far remove from the robust phenotypic inheritance originally reported in the Zeybel paper (PMID: 22941276).

I fully support acceptance of this important manuscript by NCOMMS if the following points are addressed.

- We thank the reviewer 1 for a detailed introduction and contextualization of the study. We strongly appreciate that the reviewer acknowledges the quality of the study design and analyses, highlights the importance of the results, and fully supports publication in a high-quality journal such as NCOMMS for maximum exposure and visibility.

1/ In the introduction I challenge the authors suggestion that ‘environmental experiences.....are written into our epigenomes and can be transmitted through the male and/or female germline to influence development and health of progeny.’ Outside of imprinted genes (which are a special case not involving environmental exposure, but may involve transcriptional patterning in the oocyte that lay down methylation imprints) the evidence is poor to non-existent. In this respect the Cartier (PMID 29636086) and Iqbal (ref 45 in the manuscript) papers are importance references and should be fully discussed in BOTH the introduction and discussion.

- We thank the reviewer for a careful review of the introduction. We agree that the overall evidence of germline epigenetic transmission of environmental-exposure induced phenotypes is uncertain and debated, which has been the scope of the first introduction section. A few papers had reported germline epigenetic changes underlying inherited transgenerational effects (Dias and Ressler 2014; Gapp et al. 2020; Radford et al. 2014), while others showed no significant germline changes, including the Cartier et al. and Iqbal et al. papers suggested by the reviewer (Carone et al. 2010; Iqbal et al. 2015; Cartier et al. 2018). In the revised version we have carefully reworded the introduction section and updated the citations accordingly.
- The current weight of evidence for robust transgenerational effects in mammals is further addressed in the discussion section (see also response to comment #6 from this reviewer below), resulting in the following discussion statement:
 - DISCUSSION: *“Thus, the current weight of evidence for robust transgenerational effects in mammals is weak, and the mechanistic basis by which ancestral information may be passed to progeny despite germline and zygotic waves of global epigenome reprogramming is unclear, resulting in continued debate on the optimal study design framework for generating reproducible and high-quality data.”*

2/ I applaud the authors in identifying a ghost set of 69 genes (group 11 vs group 20) that are prominent after extensive bioinformatics analysis. However, it does raise the question what would happen in other tissues from group 11 vs group 20, is there transcriptional instability present there also, given that many of the targets are not specifically expressed in hepatocytes (from a very nice single cell analysis).

- We thank the reviewer for acknowledging the identification of a “ghost” transcriptional signature. As per reviewer 2 comment below, we have now reformulated the analysis, providing the full list (n=1523) of genes under F0 treatment influence as well as the list (n=69) of genes under cumulative F0+F1 influence. We also now discuss precedents for environmental factor to influence the transgenerational epigenetic programming of an organ’s transcriptome (Skinner et

al. 2012; Guerrero-Bosagna et al. 2013; Anway and Skinner 2008; Anway, Rekow, and Skinner 2008; Manners et al. 2019) (see discussion section).

- We agree that the evaluation of transcriptional effects across tissues is relevant for a better characterization of the phenomenon. Such question and evaluation is precedented: investigation of different tissue transcriptomes in male and female F3 generation vinclozolin versus control lineage rats demonstrated all tissues examined had transgenerational transcriptomes (Skinner et al. 2012). This study suggested that “*all tissues derived from the epigenetically altered germ line develop transgenerational transcriptomes unique to the tissue, but common epigenetic control regions in the genome may coordinately regulate these tissue-specific transcriptomes.*”
- To expand on the liver transcriptome findings and based on this and reviewer 3 point #3 comments we have profiled the kidney transcriptome not only from group 11 vs 20, but from all 4 ancestral vehicle groups 11, 14, 17, 20. We selected the same 7 animals profiled for liver transcriptional effects and ran RNAseq under the same protocol as used for liver samples. In a PCA, the percent of variance in gene expression profiles captured by each PC is low ($\leq 23\%$). Excluding variance likely capturing potential effects resulting from sample processing (PC1 and PC2), we identify gene signatures in PC3 and PC4, albeit capturing only 10% and 5% of variance in the data, that separates group 11 (cohort A) animals. We next differentiated the effect of treatment in the F0 generation (FDR < 0.1) of at least moderately expressed genes (estimated base mean expression > 50) and identify 66 genes (**Supplementary Data 2**) whose expression correlates with F0 treatment history. Interestingly, 20 of those genes overlap with the F0 liver gene signature ($n=1523$). Within those we find *Zbtb16* to show the same expression distribution profile as in liver. With this additional data we cannot exclude that other tissues beyond liver may be transcriptionally affected. This additional data is reported in the results section, presented in a new **Supplementary Fig. 7** and discussed.

3/ In their bioinformatic RNA seq analysis the authors should check if there are alterations in repeat expression analysis (LINE1, IAP and ERV sequences), this relates to point 4 below. See PMID: 23043114 for an example of repeat analysis in the rat genome.

- We agree with the reviewer that this is an interesting hypothesis to follow up on, especially in the light of the reviewer’s observation in point #4. We performed a preliminary analysis of our data as discussed below towards answering the reviewer’s question. However, due to (a) the lack of an appropriate test data set to check the quality of the analysis pipeline results and (b) the use of PolyA enrichment for RNAseq library preparation, and therefore uncertainty in the interpretation of the results, we have decided to discuss the results here and not in the main manuscript.
- Ambiguously mapped reads from RNAseq, such as is the case for highly repetitive regions, are commonly disregarded from expression quantification due to high uncertainty in mapping. Analysis pipelines tailored specifically to the task of quantifying transposable elements have been published to overcome this limitation. To answer the reviewer’s point, we have applied a publicly available analysis pipeline TEtranscripts (Jin et al. 2015) to our data set. The choice to use this pipeline was based on ease of use and availability of an accompanying curated set of rat transposable elements. We applied the pipeline as is, using the rat transposable element annotation provided by the authors (<https://github.com/mhammell-laboratory/TEtranscripts>).
- On the individual transposable element level, we identified 31 LTR class elements (mostly ERVK, ERV1, ERVL-MaLR and ERVL family) and 10 DNA class elements correlating with F0 background (FDR < 0.05). Additionally, 5 SINE and 4 LINE class elements (mainly L1 family) were highlighted. The differential expression test statistics are listed in the table below. No strong correlations with F1 background became evident in the differential expression analysis (FDR < 0.2), possibly due to low overall expression in our data set (PolyA) and limited power for statistical analysis.

TE element	baseMean	log2 FoldChange	lfcSE	stat	pvalue	padj
FordPrefect:hAT-Tip100:DNA	88.92	0.9	0.205	4.38	0.000012	0.00094
LTR78:ERV1:LTR	4.27	0.76	0.243	3.11	0.0019	0.031
MamGypsy2-l:Gypsy:LTR	4.55	0.73	0.253	2.89	0.0038	0.05
7SK:RNA:RNA	74.72	0.69	0.177	3.88	0.0001	0.0041
MER92B:ERV1:LTR	286.38	0.63	0.183	3.43	0.00061	0.014
RLTR41B_Rn:ERV1:LTR	6.48	0.6	0.189	3.15	0.0016	0.029
RNERV14-int:ERVK:LTR	5.31	0.6	0.183	3.25	0.0011	0.022
NICER19A-int:ERV1:LTR	6850.74	0.59	0.115	5.12	0.00000031	0.000063
RNIAP1a:ERVK:LTR	504.5	0.58	0.184	3.17	0.0015	0.027
X1_DNA:TcMar-Tigger:DNA	8.88	0.56	0.177	3.15	0.0016	0.029
L1MDa:L1:LINE	149.9	0.55	0.163	3.36	0.00079	0.017
MTB-int:ERVL-MaLR:LTR	53.87	0.55	0.138	3.96	0.000074	0.0032
Charlie2a:hAT-Charlie:DNA	67.66	0.55	0.146	3.75	0.00018	0.006
EUTREP14:EUTREP14:Unknown	8.17	0.5	0.172	2.92	0.0036	0.047
NICER2_Rn:ERV1:LTR	1049.87	0.49	0.104	4.69	0.0000028	0.00032
RLTR18:ERVK:LTR	205.37	0.47	0.142	3.32	0.00089	0.019
RLTR12C:ERVK:LTR	35.65	0.44	0.125	3.54	0.00039	0.011
MER91B:hAT-Tip100:DNA	55.22	0.41	0.121	3.39	0.00071	0.016
MER91A:hAT-Tip100:DNA	80.73	0.41	0.098	4.21	0.000026	0.0016
RNERVK8d-int:ERVK:LTR	78.49	0.41	0.141	2.93	0.0034	0.046
L1M6:L1:LINE	45.39	0.39	0.103	3.79	0.00015	0.0054
MER45A:hAT-Tip100:DNA	20.39	0.39	0.129	3.03	0.0025	0.037
Tigger1:TcMar-Tigger:DNA	62.84	0.39	0.094	4.17	0.00003	0.0018
RLTR22_Rat1:ERVK:LTR	72.49	0.36	0.124	2.89	0.0038	0.049
RNLTR8C:ERVK:LTR	26	0.35	0.115	3.02	0.0025	0.038
Plat_L3:CR1:LINE	55.25	0.34	0.1	3.34	0.00083	0.018
RNLTR7-int:ERV1:LTR	301.14	0.34	0.114	2.96	0.0031	0.043
RMER17C-int:ERVK:LTR	36.11	0.34	0.104	3.23	0.0012	0.024
RMER3B:ERVK:LTR	1316.61	0.34	0.107	3.17	0.0015	0.027
RNERVK8c-int:ERVK:LTR	313.78	0.34	0.106	3.23	0.0012	0.024
MLTR11A:ERVK:LTR	223.89	0.34	0.08	4.27	0.000019	0.0013
RNERVK23-int:ERVK:LTR	200.61	0.33	0.076	4.33	0.000015	0.0011
RNLTR23:ERVK:LTR	46.35	0.33	0.098	3.4	0.00067	0.015
MER53:hAT:DNA	171.02	0.31	0.082	3.85	0.00012	0.0045
MLT1F:ERVL-MaLR:LTR	232.45	0.28	0.096	2.96	0.0031	0.043
MTB:ERVL-MaLR:LTR	172.26	0.27	0.085	3.13	0.0017	0.03
RMER1A:RMER1A:Other	254.91	0.26	0.074	3.46	0.00055	0.014
MLT1A0:ERVL-MaLR:LTR	303.66	0.26	0.082	3.19	0.0014	0.026
RNLTR16B:ERV1:LTR	61.3	0.24	0.059	4.01	0.00006	0.0028
RNLTR4d_I:ERVK:LTR	252.83	0.23	0.065	3.57	0.00036	0.01
Charlie8:hAT-Charlie:DNA	122.31	0.23	0.074	3.16	0.0016	0.028
RMER15:ERVL:LTR	700.87	0.22	0.063	3.44	0.00059	0.014
Lx9:L1:LINE	2627.81	0.22	0.071	3.1	0.002	0.032
B2_Rat3:B2:SINE	1298.54	0.2	0.067	2.93	0.0034	0.046
ID_Rn1:ID:SINE	2155.87	0.19	0.052	3.65	0.00027	0.008

MER21C:ERV1:LTR	374.7	0.19	0.056	3.44	0.00059	0.014
MTE2b-int:ERV1-MaLR:LTR	182.5	0.18	0.06	3.03	0.0025	0.037
RMER5:ERV1:LTR	412.73	0.17	0.056	3	0.0027	0.039
RSINE1:B4:SINE	10109.09	0.17	0.057	2.99	0.0028	0.04
MIRb:MIR:SINE	2876.64	0.17	0.057	3.04	0.0023	0.036
ID_Rn2:ID:SINE	3197.14	0.14	0.044	3.14	0.0017	0.029
LTR83:ERV1:LTR	24.47	-0.3	0.1	-3.02	0.0026	0.038
MER99:hAT:DNA	8.02	-0.61	0.202	-3.01	0.0027	0.039

- For all except two identified Transposable Elements (TE), the fold-changes indicate higher expression if F0 generations have been exposed to treatment (see examples below). Indeed, lowest expression for many of the identified TE elements is observed when both F0 and F1 generations were not exposed to treatment. It is tempting to speculate relevance of the mechanism described in reviewer's point #4, as the described pattern mirrors the *Lsh/Hells* gene expression across groups. However, further careful evaluation of RNA, protein and methylation levels would be necessary to evaluate if an underlying mechanism resembles the one previously described for parental irradiation in the publication mentioned by the reviewer (Filkowski et al. 2010), and if an effect on *Lsh/Hells* and LINE and SINE B2 elements is at play in the context of our liver injury model.

4/ The authors seemed to have missed a trick in not highlighting one of the factors that they identified in the ghost set, notably *Lsh/Hells*, which they have previously reported on being upregulated in response to Phenobarbital exposure in the liver, which should be referenced (PMID: 24690595). This is of great interest because *Hells* is an epigenetic co-regulator for global DNA methylation levels and is required to maintain repression of IAP LTR and satellite repeat expression in mouse somatic cells (PMID: 24367978). Its mis-regulation is also linked with epigenetic changes in the male germline after paternal irradiation, which results in deleterious effects in the somatic thymus tissue from the progeny of exposed animals (PMID: 19959559). These observations are highly relevant to their study and should be discussed fully.

- We agree with the reviewer that *Lsh/Hells* represents one functionally interesting gene that could be highlighted to the reader and relates to point #3 above. In the revised version of the manuscript, and responding to the reviewer 2 comment below, we now report both the full list of genes under F0 treatment history (1523 genes) and cumulative F0+F1 treatment history (69 genes), supporting transgenerational transmission at the molecular level. GO term and individual gene assessment point to potential changes in chromatin modifying genes. Through its reported roles as a chromatin co-effector, regulation of transposable elements and function in male germline epigenetic changes, *Lsh/Hells* represents a potentially interesting molecular and functional integrator. We have modified the results and discussion sections to exemplify the potential biological relevance of this gene set.
 - RESULTS: *Consistent with the enrichment for genes involved in chromatin modifications among the 1523 genes correlated with F0 treatment history, we also find Hells (a.k.a Lsh or Smarca6), a member of the SNF2 DNA helicase family and important cell proliferation regulator involved in chromatin modifications and structuration among the upregulated genes in this F0+F1 gene set (Supplementary Fig. 6).*
 - DISCUSSION: *Interestingly, the altered expression of epigenetic (co)effectors such as histone and DNA demethylases or such as Hells/Lsh provides the possibility that epigenetic processes could be involved. Hells is directly required for the methylation and silencing of transposable repetitive elements during gametogenesis and in somatic cells and its mis-regulation was linked to male germline epigenetic changes following paternal irradiation, resulting in deleterious effects in the somatic thymus tissue from the progeny of exposed animals.*
- Importantly though, detailed phenotypic investigations in this study do not support adaptive phenotypic suppression of the hepatic wound-healing response as there was no evidence of reduced liver fibrogenesis or greater fitness of animals with ancestral liver injury. In the absence of detectable phenotypic effects, the functional relevance of this hepatic transcriptional programming or associated transmission mechanisms is unclear and would warrant further investigations. Our work overall highlights the need for further evaluation of transgenerational epigenetic inheritance paradigms in mammals.

5/ I note that the labelling for *Slc10a2* is cut off in figure 5b.

- Thank you for carefully reviewing the figures. We have corrected the figure accordingly, adapted all submitted images to high resolution versions.

6/ I think the discussion should be more robust in scope, its actually good news if TGI due to environmental exposure appears to be a minor phenomenon in animals at best. As in the CCl4 exposure model, initial observations have not been completely validated. Here, the Cartier and Iqbal studies should be fully discussed and highlighted, it is interesting that the phenotypic consequences of prenatal glucocorticoid overexposure is well documented (<https://pubmed.ncbi.nlm.nih.gov/?term=prenatal+glucocorticoid+overexposure&sort=pubdate>). But for

the glucocorticoid rat model in the F2 generation, although birth weight is also reduced, the effects on foeto-placental growth and gene expression differed from those in the F1, with marked parent of origin effects (PMID: 22086116). This implies that the mechanisms underlying glucocorticoid programmed effects in F1 and F2 generations differ. Notably the glucocorticoid-programmed phenotype did not persist into a third generation (F3) (PMID: 15178540). There were no differences between F3 offspring groups in birth weight, litter number, or gestation length or postnatal growth patterns, which is also good news and should be discussed.

- We agree with reviewer 1 that the discussion could be extended to bring further perspective to the lack of reproducibility of the originally reported transgenerational phenotypic adaptation to ancestral liver injury. This comment is also in line with comments from reviewer 2 and 3, including the suggestion to adapt the title to better carry the key message of unaltered fibrosis (reviewer 3 point #7). We have thus revised the discussion to highlight the debate around transgenerational phenotypic effects and the lack of understanding of the mechanistic basis for (germline epigenetic) transmission.
- We specifically discuss and reference a couple of reported models of multigenerational responses (vinclozolin and glucocorticoid overexposure) to illustrate the debate around the overall evidence for transgenerational effects in mammals and the mechanistic basis by which ancestral information may be passed to progeny despite germline and zygotic waves of global epigenome reprogramming. Notably, we differentiate '*transgenerational*' effects (found in generations that were not exposed to the initial signal or environment that triggered the change) from '*intergenerational*' effects (such as the direct *in utero* impact of the developing embryo and its germline (which will eventually produce grandchildren), as defined in various papers including by (Heard and Martienssen 2014).

If a revised version of this manuscript is accepted by NCOMMS, it certainly deserves an accompanying news and views comment highlighting its significance.

- We thank the reviewer for acknowledging the importance of our submitted manuscript and advocating a news & views highlight of its significance.

Reviewer #2 (Remarks to the Author):

In this study, the authors performed an extremely well-controlled study of the inter-generational effects of CCL4 exposure in rats, repeating a previous study that showed heritable morphological phenotypic effects on liver fibrosis response. The morphological effects were not repeated, but gene expression effects were apparent across generations, that correlated with direct and ancestral treatments. The methods are, for the most part, appropriate, and gene expression results were validated by several methods. The overall results are sound.

This is an important paper, even though a portion of it gives 'un-sexy' negative results. The field of heritable epigenetic effects on gene expression has suffered from criticism that controls, validation and experimental design are inadequate, and this study addresses those concerns well.

- We thank the reviewer for reinforcing the importance of our independent assessment of the reported transgenerational effects and for acknowledging the soundness of our study design and results, necessary to manage the complexity of factors influencing transgenerational effects.

I have only one concern: in Lines 269-271 the authors state “We assumed that F0, F1 and F2 effects on modulating gene expression in the F2 generation are additive, i.e., resulting in cumulative changes across generations.” I do not think that this is a valid assumption, as epigenetic and gene expression changes are seen to be quite different in F1, F2 and F3 generations in previous rat studies. It might be better to not limit the evaluated dataset with this parameter.

- We agree with the reviewer that not all genes affected by F0 treatment follow the assumption of additive effects across generations. This was also referenced in point #6 by reviewer 1, describing differences between glucocorticoid programmed effects in F1 and F2 generations (Drake et al. 2011).
- We have adapted the results section and data reporting through reporting both the full list of genes under F0 treatment history (1523 genes) and cumulative F0+F1 treatment history (69 genes) in stepwise manner:
 - RESULTS: *We first differentiated the effect of treatment in the F0 generation (FDR <0.1) of at least moderately expressed genes (estimated base mean expression >50). This resulted in 1523 genes correlated with F0 treatment history (Supplementary Data 2)*
 - RESULTS: [...] *We thus expanded gene expression analysis to consider potentially additive F0, F1 effects on modulating gene expression in F2. We focused on genes showing the same log2 fold-change direction for the estimated F1 treatment effect, resulting in 1304 genes with additive F0 and F1 treatment effects. We further assumed that with cumulative effects, transcriptional effects should be most apparent when comparing the control groups of cohort A vs D, and to a lesser extend of cohort A vs C (as underlined by the observations based on the PCA in Fig. 45a), narrowing down the selection to 69 genes that show FDR <0.1 in the group 11 vs group 20 contrast and FDR <0.2 in the group 11 vs group 17 contrast (Supplementary Data 2).*
- For the sake of orthogonal validation, we maintained focus on the cumulative F0+F1 effect gene set (69 genes) which represents a more manageable set of genes for additional experimental validation of RNAseq derived gene expression changes. We highlight a few additional genes and functional GO terms and maintain the orthogonal evaluation in the last figure.
- The full transcriptional data are available under GEO accession number GSE229524 for further analyses.

A very interesting and important study for those interested in the emerging field of neo-Lamarckian heritability.

- We thank the reviewer for acknowledging the importance of our submitted manuscript and significance for the field, overall highlighting the need for further evaluation of transgenerational epigenetic inheritance paradigms in mammals.

Reviewer #3 (Remarks to the Author):

Beil et al present a carefully designed and performed multigenerational study on the effects of ancestral CCl4 injury, with increased power, full pedigree tracing, randomization scheme and an F2 dose-response evaluation. Their data do not support adaptive phenotypic suppression of the hepatic wound-healing response or a greater fitness of animals with ancestral liver injury exposure as originally reported. The authors suggest that there is an influence of ancestral injury on hepatic gene expression, but this part of the study remains vague, statistically borderline and without clear interpretation/guidance:

Comments:

1. The transgenerational study is carefully performed with sufficiently large enough cohorts and corrects a previous study that includes only small cohorts of rats (n=5).

- We thank the reviewer for acknowledging the quality of our study design, including large enough cohorts. Our design considered additional important parameters as summarized in the results section. In brief: 1. standardized housing and care conditions. 2. Study power (cohort size) in all 3 generations. 3. Pedigree representation / information. 4. F2 dose-response evaluation. 5. Carefully designed staggering and randomization scheme for F2 dose-response treatment, collection, and evaluation. 6. Comprehensive sampling and phenotypic. A detailed Supplementary Methods document is available.

2. The transcriptomics studies remain vague. While the authors point out transcriptomic differences that they describe as “associated with molecular pathways underlying liver homeostasis and fibrosis”, the GO terms in Fig.4D are not directly linked to fibrosis or liver homeostasis, many of them very broad and not linked to fibrosis or homeostasis or sounding vaguely like ECM or fibrosis but not really linked to fibrosis (e.g. regulation of cell-matrix adhesion, regulation of cell junction, , focal adhesion....). Likewise, the genes shown in Fig.4 and 5 are not well known central players in homeostasis or fibrosis.

- Our study primarily shows that we could not reproduce the morphological effects of liver fibrosis adaptation initially reported (Zeybel et al. 2012). Given the lack of a strong transgenerational effect on fibrosis phenotype, we have taken an unsupervised RNAseq based approach to gene expression analysis. We report transcriptional changes potentially related to ancestral treatment, independent of F2 fibrotic state. This ancestral influence is primarily detected from F2 vehicle and low dose groups, plausibly in conditions of lower degree of tissue injury and associated transcriptional perturbations.
- In the revised manuscript and based on this and reviewer 2 comments, we have now reformulated the analysis, providing the full list (n=1523) of genes under F0 treatment influence as well as the list (n=69) of genes under cumulative F0+F1 influence. We provide GO term enrichments analysis of both sets of genes.
 - The GO term enrichment analysis on F0 treatment correlated 1523 genes highlight perturbation of biological pathways related to ribosome function, chromatin modifications (including a broad range of histone demethylases and DNA methylation regulating genes such as *Dnmt3b* and *Tet* family genes), as well as tissue developmental and differentiation pathways (including *WNT/b-catenin*, *TGF- β* and *Notch1* signaling pathways).
 - As initially reported, the GO term enrichment analysis on cumulative F0+F1 treatment correlated 69 genes identified weak enrichment for genes associated with protein deacetylation, cell-substrate junction, cell-matrix and focal adhesion, hemopoiesis, as well as regulation of fat cell differentiation.
- To increase our confidence in the RNAseq derived gene expression changes we have orthogonally evaluated a subset of genes (using RT-qPCR and ISH) and find strong inter-sample and cross-assay correlation (**Supplementary Fig. 8**), supporting the robustness of the detected changes, for the selected exemplary genes.
- While intriguing to speculate relevance to transgenerational inheritance, we agree with the reviewer that the functional and phenotypic consequences of the identified gene signature remain unclear because we could not anchor them to any transgenerational effects of fibrotic response in our study.
- Interestingly, the ability of an environmental factor to influence the transgenerational epigenetic programming of an organ's transcriptome is precedented and we now provide a more detailed description of literature supporting this phenomenon in the discussion section.

Most importantly, many of the data that are shown are not statistically significant. I would suggest to use strict statistical criteria such as $FDR < 0.05$ for RNA-seq and $p\text{-values} < 0.05$ for single genes. If one then focuses on homeostasis-relevant genes, literally nothing remains. Unless the authors have functional studies where they can prove the relevance of these genes. To the reviewer's eyes, this looks like a desperate attempt to rescue the transgenerational concept or to provide some data to appease authors from previous publications. The authors need to show some hard-core data what the transgenerational difference and their meaning are. This is not convincing and could be due to chance given the low significance.

- Given the small variance in gene expression observed across F2 control groups in a PCA analysis on RNAseq data (new **Fig 5a**), we have decided to apply a FDR of 0.1 as cut-off for selecting differentially expressed genes for the relevant contrasts (F0-related gene expression changes). We thereby accept a larger percentage of expected false positives than with a cutoff of $FDR < 0.05$. Following reviewer 2 and 3's comments, we now provide the full list of $n=1523$ F0-related genes (based on FDR cut-off 0.1 and base mean expression cut-off of 50) in **Supplementary Data 2**.
- We performed several additional investigations to assess the relevance of the selected genes. We could identify enriched biological pathways that show coherence in biological function for the selected genes (new Supplementary **Fig 5** and **Fig 5d**). Additionally, we selected a subset of genes for further validation by orthogonal assays, guided by a comprehensive bioinformatic analysis of the initial RNAseq derived dataset as described in the main text. Further support for the relevance of the selected genes was obtained through a detailed literature survey on their potential roles in liver homeostasis.
- However, as rightly pointed out by the reviewer, the gene signature's role in any protective effect to liver injury in F2 remains elusive, especially given the absence of microscopic or clinical pathology-based evidence for multigenerational adaptation of the hepatic wound-healing response in our study. We discuss experimental factors potentially leading to the lack of F2 phenotypic outcome in the discussion.

3. The concept of ancestral CCL4 injury exclusively affecting the liver is not convincing. The reviewer is aware of the previous Nat Med paper, but the authors should check the concept that the transgenerational effects could be on other organs as well, as effects are transmitted through the germline rather than the hepatocyte or other liver cells. As the authors preserved additional organs, it would be interesting to run one additional RNA-seq in another organ and compare differences in gene expression. This could be simply the untreated groups 11,14,17 and 20.

- As discussed above in response to reviewer 1 point #2:
 - We agree that the evaluation of transcriptional effects across tissues is relevant for a better characterization of the phenomenon. Such question and evaluation is precedented: investigation of different tissue transcriptomes in male and female F3 generation vinclozolin versus control lineage rats demonstrated all tissues examined had transgenerational transcriptomes (Skinner et al. 2012). This study suggested that "*all tissues derived from the epigenetically altered germ line develop transgenerational transcriptomes unique to the tissue, but common epigenetic control regions in the genome may coordinately regulate these tissue-specific transcriptomes.*"
 - To expand on the liver transcriptome findings and based on this and reviewer 3 point #3 comments we have profiled the kidney transcriptome not only from group 11 vs 20, but from all 4 ancestral vehicle groups 11, 14, 17, 20. We selected the same 7 animals profiled for liver transcriptional effects and ran RNAseq under the same protocol as used for liver samples. In a PCA, the percent of variance in gene expression profiles captured by each PC is low ($\leq 23\%$). Excluding variance likely capturing potential effects resulting

from sample processing (PC1 and PC2), we identify gene signatures in PC3 and PC4, albeit capturing only 10% and 5% of variance in the data, that separates group 11 (cohort A) animals. We next differentiated the effect of treatment in the F0 generation (FDR <0.1) of at least moderately expressed genes (estimated base mean expression >50) and identify 66 genes (**Supplementary Data 2**) whose expression correlates with F0 treatment history. Interestingly, 20 of those genes overlap with the F0 liver gene signature (n=1523). Within those we find *Zbtb16* to show the same expression distribution profile as in liver. With this additional data we cannot exclude that other tissues beyond liver may be transcriptionally affected. This additional data is reported in the results section, presented in an additional **Supplementary Figure 7** and discussed.

4. It is understood that the authors follow some of the previous groups in their protocols, but it would have been more convincing to have longer treatments of CCl₄ in F0 and F1 to allow for stronger potential transgenerational effects.

- Thank you for highlighting alternative or additional study design parameters that could influence the strength of the transgenerational effects.
- As the reviewer points out the primary aim of this study has been to independently evaluate the “multigenerational epigenetic adaptation of the hepatic wound-healing response” phenomenon originally published in Nature Medicine by Zeybel et al. 2012, using a carbon tetrachloride (CCl₄)-induced liver injury model and male germline transmission paradigm. We have thus followed the same basic paradigm, especially with regards to treatment and recovery durations.
- We thus repeated the original evaluation in a fully independent three-generation in vivo rat study conducted in compliance with the Animal Welfare Act, the Guide for the Care and Use of Laboratory Animals, and the Office of Laboratory Animal Welfare. In extending the original study, we carefully considered important study features such as animal house and care, high study power, cross-generational pedigree tracing and representation and careful staggering and randomization schemes for F2 dose-response treatment, collection, and evaluation. We consider that the study was run to very high technical and scientific standards and represents one of the best-in-class studies evaluating this phenomenon in a rodent setting.
- Further extended evaluation of this paradigm (e.g. to consolidate or phenotypically explore the relevance of the inherited transcriptional gene signature) could include extended treatment durations, varied lengths of inter-generational recovery, the potential for maternal transmission and the maintenance and impact of effects in further generations. However, these explorations are beyond the scope of the current manuscript and would require unprecedented resources that likely would require support from a future public-private partnership.

5. Not to beat a dead horse, but since this is a mostly negative study, it would be helpful to confirm the histological fibrosis in Figure 2b by one simple additional test, α SMA IHC and morphometric quantification. This works very well and will help to convince the more critical reviewers. Likewise, it would be helpful and very easy to add more fibrogenic genes including Col1a2, Acta2, Lox and TIMP1 (with existing material/cDNA, a matter of a few days to get the data).

- The primary objective of this study was to robustly evaluate the potential for transgenerational adaptation of the hepatic wound-healing response in a de novo multi-generation CCl₄ rat study. The previously reported morphological effects were not reproduced and we agree that an even broader range of relevant evaluations and assays would further support our conclusions.
- As per the reviewer’s suggestion, we have evaluated α SMA in all F2 study samples. We have also evaluated a few more fibrogenic genes using RT-qPCR. No ancestral influence was detected for any of the new endpoint evaluated, in any of the dose group. Notably, the RT-qPCR evaluation and the unsupervised RNAseq based transcriptome evaluation are strongly correlated (compare data outcome from **Fig. 2, Fig. 4, Fig. 6, Supplementary Fig. 6 and 8**).

- We have included the additional data in the results section as follows:
 - RESULTS: [...] *“Likewise, we could not detect lower gene expression of hepatic collagen I (Col1a1 and Col1a2) or other liver fibrosis-promoting genes such as Lox 25 or Timp1 26 in any of the dose groups evaluated (Fig. 2c).”*
 - RESULTS [...] *To further confirm the histological fibrosis and orthogonally evaluate potential cross-generational effects, we next assessed alpha smooth muscle actin (α SMA), a marker of myofibroblasts, which are important drivers of liver fibrosis. Anti- α SMA IHC staining of FFPE liver sections revealed a dose-dependent increase in myofibroblasts in all generational cohorts (Fig. 3a, c). No apparent ancestral exposure effects were observed in either dose group as illustrated in Fig. 3b and quantified in Fig. 3c. Likewise, the expression of the α SMA encoding gene Acta2 while influenced by treatment did not further detect ancestral influence for this fibrotic marker (Fig. 3d).*
- In addition, while we had focused the H&E pathology evaluation on high dose and vehicle groups in the original submission, in this revised version, the pathology evaluation was extended to include the low dose group (**Supplementary Data1**). Low dose CCl₄ treated groups 12, 15, 18 and 21 showed mild liver changes (degenerative and fibrotic), affecting in similar manner all animals, regardless of the ancestral CCl₄ exposure.

6. Since regeneration and inflammation are an important aspect for outcomes and inevitably follow injury, I would also suggest to determine proliferation by Ki67 IHC, CD45 IHC and qPCR for a few select genes.

- We agree that regeneration and inflammation are relevant aspect for outcomes and inevitably follow injury. Based on our unsupervised transcriptome and GO term enrichment analyses, we did not detect gross differences in liver injury and associated proliferative or inflammatory response. A new statement is provided in the results section.
 - RESULTS: *“Comparison of the selected differentially expressed genes showed general overlap in molecular changes (Fig. 4e) and of enriched GO terms across cohort A-D treatment groups (Supplementary Fig. 4i), including biological functions related to extra-cellular matrix, cellular proliferation (e.g., cell division, nuclear division), and inflammation (e.g. leukocyte chemotaxis, cytokine production and T-cell activation). Thus, no gross differences in liver injury and associated proliferative or inflammatory response were detected.”*
- The RNA sequencing data is publicly available under the GEO accession number GSE229524 for any further gene or gene-set analyses.

7/ Depending on where this is heading, the authors may need to change the title. The current title is weak as it is not backed by strong findings and as it does not carry the key message of unaltered fibrosis.

- We agree with reviewer 3 and thank him for this important suggestion. As equally highlighted by reviewers 1 and 2, despite the characterization of inherited hepatic transcriptional changes, the most impactful outcome of our study remains that we were unable to detect adaptive phenotypic suppression of the hepatic wound healing response or a greater fitness of animals with ancestral liver injury exposure, mitigating the impact of the original 2012 report.
- We thus adapt our manuscript title to **“Unaltered hepatic wound-healing response in male rats with ancestral liver injury”**. This new title captures the wound healing in general without focusing on fibrosis only (as in the original report), highlights the model and focus on male evaluation, and brings the concept of transgenerational changes or the lack thereof.

8. With all above, the authors should also carefully revisit whether there is possibly a lower degree of

fibrosis/inflammation/proliferation in the group 12 vs 15, 18 or 21 (which would be the opposite of what was reported in the 2012 Nat Med paper).

- The primary objective of this study was to robustly evaluate the potential for transgenerational adaptation of the hepatic wound-healing response in a *de novo* multi-generation CCl₄ rat study. In this study, we expanded from the original evaluation through considering additional study parameters that mitigate potentially confounding effects. In doing so, we evaluated F2 animals not only at high dose (as in the original Zeybel et al 2012 study) but also with control and low dose treatment, which is a significant addition to the original study. The analysis of high dose samples does not support adaptive phenotypic suppression of the hepatic wound-healing response as there was no evidence of reduced liver fibrogenesis or greater fitness of animals with ancestral liver injury.
- To the reviewer's point, in the low dose group, small trends (group 12 vs 15, 18, 21) may be visually noted in a few graphs: lower degree of Sirius Red (**Fig 2a**), higher degree of hepatocyte degeneration (**Fig 2f**), which might point to ancestral influence. However, none of those quantitative imaging trends reach statistical significance.
- As mentioned in the point #5 above, while we had originally focused the H&E pathology evaluation on high dose and vehicle groups, we now extended the evaluation to include low dose group alongside previous data (**Supplementary Data 1**). Low dose CCl₄ treated groups 12, 15, 18 and 21 showed mild liver changes (degenerative and fibrotic), affecting in similar manner all animals, regardless of the ancestral CCl₄ exposure. Thus, no obvious effect or directionality thereof can be claimed from this data.
- Our control and low dose transcriptome analyses do however support transgenerational transmission at the molecular level and GO term enrichment analysis points to gene sets of potential biological relevance. As discussed, while intriguing to speculate relevance of these transcriptomic changes for transgenerational inheritance, the functional and phenotypic consequences of the identified gene signatures remain unclear because we could not anchor them to any transgenerational effects of fibrotic response in our study.

REFERENCES

- Anway, M. D. & Skinner, M. K. Transgenerational effects of the endocrine disruptor vinclozolin on the prostate transcriptome and adult onset disease. *Prostate* 68, 517–29 (2008).
- Anway, M. D., Rekow, S. S. & Skinner, M. K. Transgenerational epigenetic programming of the embryonic testis transcriptome. *Genomics* 91, 30–40 (2008).
- Carone, B. R. et al. Paternally induced transgenerational environmental reprogramming of metabolic gene expression in mammals. *Cell* 143, 1084–96 (2010).
- Cartier, J. et al. Investigation into the role of the germline epigenome in the transmission of glucocorticoid-programmed effects across generations. *Genome Biol* 19, 50 (2018).
- Dias, B. G. & Ressler, K. J. Parental olfactory experience influences behavior and neural structure in subsequent generations. *Nat Neurosci* 17, 89–96 (2014).
- Drake, A. J., Liu, L., Kerrigan, D., Meehan, R. R. & Seckl, J. R. Multigenerational programming in the glucocorticoid programmed rat is associated with generation-specific and parent of origin effects. *Epigenetics* 6, 1334–43 (2011).
- Filkowski, J. N. et al. Hypomethylation and genome instability in the germline of exposed parents and their progeny is associated with altered miRNA expression. *Carcinogenesis* 31, 1110–5 (2010).
- Gapp, K. et al. Alterations in sperm long RNA contribute to the epigenetic inheritance of the effects of postnatal trauma. *Mol Psychiatry* 25, 2162–2174 (2020).
- Guerrero-Bosagna, C., Savenkova, M., Haque, M. M., Nilsson, E. & Skinner, M. K. Environmentally induced epigenetic transgenerational inheritance of altered Sertoli cell transcriptome and epigenome: molecular etiology of male infertility. *PLoS One* 8, e59922 (2013).
- Heard, E. & Martienssen, R. A. Transgenerational epigenetic inheritance: myths and mechanisms. *Cell* 157, 95–109 (2014).
- Iqbal, K. et al. Deleterious effects of endocrine disruptors are corrected in the mammalian germline by epigenome reprogramming. *Genome Biol* 16, 59 (2015).
- Jin Y, Tam O. H., Paniagua E, Hammell M. TETranscripts: a package for including transposable elements in differential expression analysis of RNA-seq datasets. *Bioinformatics* 31(22) 3593-9 (2015)
- Jin, X., Aimaiti, Y., Chen, Z., Wang, W. & Li, D. Hepatic stellate cells promote angiogenesis via the TGF- β 1-Jagged1/VEGFA axis. *Exp Cell Res* 373, 34–43 (2018).
- Manners, M. T. et al. Transgenerational inheritance of chronic adolescent stress: Effects of stress response and the amygdala transcriptome. *Genes Brain Behav* 18, e12493 (2019).
- Radford, E. J. et al. In utero effects. In utero undernourishment perturbs the adult sperm methylome and intergenerational metabolism. *Science* 345, 1255903 (2014).
- Skinner, M. K., Manikkam, M., Haque, M. M., Zhang, B. & Savenkova, M. I. Epigenetic transgenerational inheritance of somatic transcriptomes and epigenetic control regions. *Genome Biol* 13, R91 (2012).

REVIEWERS' COMMENTS

Reviewer #1 (Remarks to the Author):

I have read the revised version of the manuscript several times, and congratulate the authors for their comprehensive responses to the reviewer's queries, the vastly improved manuscript and their important findings. I fully support publication of this manuscript in Nature Comms.

Reviewer #2 (Remarks to the Author):

The authors have addressed well all of my previous concerns as a reviewer. I recommend publication.

Reviewer #3 (Remarks to the Author):

The authors have been responsive to the previous comments and the study provides important data that challenges the concept of transgenerational adaptation and greater fitness.

The relevance of the observed transcriptomic alteration and the concept of "transgenerational transmission at the molecular level" remain the weakness of the paper as there are no clear functional consequences and as this kind of contradicts the paper's title. These studies would require further analyses - either the transcriptomic alterations are very minor and not relevant or they have a relevance that needs to be uncovered.

REVIEWER COMMENTS – Final Revision - Beil, Pfaller, Perner et al. NCOMMS-22-45904A

Reviewer #1 (Remarks to the Author):

I have read the revised version of the manuscript several times, and congratulate the authors for their comprehensive responses to the reviewer's queries, the vastly improved manuscript and their important findings. I fully support publication of this manuscript in Nature Comms.

- We thank the reviewer 1 for his/her thorough review of the revised manuscript and response to reviewer's queries. We agree the overall edits enabled strong improvement of the manuscript. We also thank the review for strong support for the publication of this study in Nat. Communications.

Reviewer #2 (Remarks to the Author):

The authors have addressed well all of my previous concerns as a reviewer. I recommend publication.

- We thank the reviewer 2 for critical assessment of the revised version of the manuscript and for continuous support for the publication of this study in Nat. Communications.

Reviewer #3 (Remarks to the Author):

The authors have been responsive to the previous comments and the study provides important data that challenges the concept of transgenerational adaptation and greater fitness.

The relevance of the observed transcriptomic alteration and the concept of "transgenerational transmission at the molecular level" remain the weakness of the paper as there are no clear functional consequences and as this kind of contradicts the paper's title. These studies would require further analyses - either the transcriptomic alterations are very minor and not relevant or they have a relevance that needs to be uncovered.

- We thank the reviewer 3 for acknowledging that this study represents important development in the field. The reviewer also highlights that the concept of "transgenerational transmission at the molecular level" is not supported by functional phenotypic data and that these studies would require further analysis.
- As discussed in the original response to reviewer's comments, while intriguing to speculate relevance to transgenerational inheritance, we did agree with the reviewer that the functional and phenotypic consequences of the identified gene signature remain unclear because we could not anchor them to any transgenerational effects of fibrotic response in our study. We believe that we had captured this limitation with necessary precautions in the text. However, acknowledging this limitation further and in alignment with the editors we have rephrased a few statements in the abstract, introduction and discussion. We hope those editorial changes appropriately address the limitation.
 - **Abstract:** *However, transcriptomic analyses identified genes whose expression correlates with ancestral liver injury, although the biological relevance of this apparent transgenerational transmission at the molecular level changes remains to be determined.*
 - **Introduction:** *Instead of: "we identify a biologically relevant set of genes whose expression pattern is under apparent influence of ancestral CCl4-induced liver injury, supporting transgenerational transmission at the*

molecular level.” We have modified to “we identify a biologically relevant set of genes whose expression pattern correlates with ancestral CCl4-induced liver injury, suggesting transgenerational transmission at the molecular level.”

- **Discussion:** *The functional relevance of the identified gene sets on liver fibrosis adaptation and the nature and transmission of potential germline and somatic (epi-)genomic changes remain unclear and require further investigations.” and further down “...the biological significance of the apparent transcriptional transmission and uncertainty on underlying mechanisms limit definitive conclusions.”*